# QC-StyleGAN - Quality Controllable Image Generation and Manipulation

Dat Viet Thanh Nguyen[1,*]     Phong Tran[1,2,*]     Tan M. Dinh[1]

Anh Tuan Tran[1]     Cuong Pham[1,3]

[1]VinAI Research    [2]MBZUAI    [3]Posts & Telecommunications Institute of Technology

`{v.datnvt2, v.tandm3, v.anhtt152, v.cuongpv11}@vinai.io`    `the.tran@mbzuai.ac.ae`

## Abstract

The introduction of high-quality image generation models, particularly the Style-GAN family, provides a powerful tool to synthesize and manipulate images. However, existing models are built upon high-quality (HQ) data as desired outputs, making them unfit for in-the-wild low-quality (LQ) images, which are common inputs for manipulation. In this work, we bridge this gap by proposing a novel GAN structure that allows for generating images with controllable quality. The network can synthesize various image degradation and restore the sharp image via a quality control code. Our proposed QC-StyleGAN can directly edit LQ images without altering their quality by applying GAN inversion and manipulation techniques. It also provides for free an image restoration solution that can handle various degradations, including noise, blur, compression artifacts, and their mixtures. Finally, we demonstrate numerous other applications such as image degradation synthesis, transfer, and interpolation.

## 1 Introduction

Image generation has achieved a marvelous development in recent years thanks to the introduction of Generative Adversarial Networks (GAN). StyleGAN models [1, 2, 3, 4] manage to generate realistic-looking images with the resolution up to 1024×1024. Their synthetic images of non-existing people/objects can fool human eyes [5, 6]. The development of such high-quality synthesis models has also introduced a new direction to effectively solve the image manipulation tasks, which usually first fit the input image to the model's latent space via a GAN "inversion" technique [7, 8, 9, 10, 11, 12, 13], then apply a learned editing [14, 15, 16] on the fitted latent code for the desired change on the generated image. Their impressive manipulation results promise various practical applications in entertainment, design, art, and more.

While the recent GAN models, notably the StyleGAN series, show promising image editing performance, we argue that image quality is an Achilles' heel, making it challenging for them to work on real-world images. StyleGAN models are often trained from sharp and high-resolution images as the desired output quality. In contrast, in-the-wild images might have various qualities depending on the capturing and storing conditions. Many degradations could affect these images, including noises, blur, downsampling, or compression artifacts. They make GAN inversion hard, sometimes impossible, to fit the low-quality inputs to the high-quality image domain modelled by StyleGAN generators. Incorrect inversions might lead to unsatisfactory editing results with obvious content mismatches. For example, a popular StyleGAN-based image super-resolution method [17] caused a controversy by approximating a high-resolution picture of Barack Obama as an image of a white man.

---

*authors contributed equally

36th Conference on Neural Information Processing Systems (NeurIPS 2022).

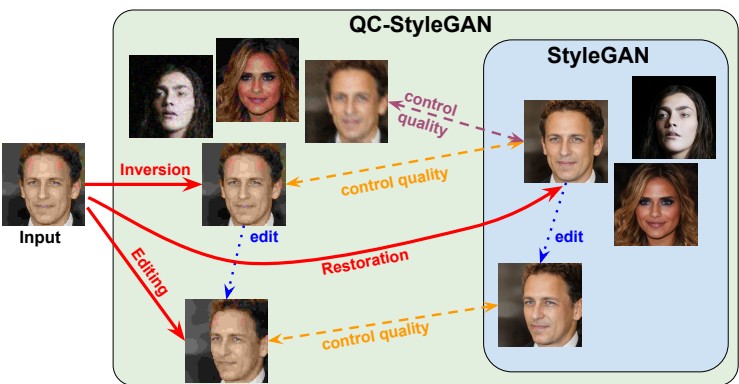

Figure 1: Our QC-StyleGAN allows for synthesizing sharp images, similar to the standard StyleGAN, and degraded images. It provides a quality-control input for easy conversion between degraded images and their sharp versions (dashed arrows). The same quality codes produce the same degradation (yellow arrows), and QC-StyleGAN covers a wide range of degradations (yellow vs. magenta). We can easily edit degraded images using the editing directions learned for sharp images in StyleGAN space (blue dotted arrows). Given a low-quality input, QC-StyleGAN allows more accurate GAN inversion, direct image editing with quality preserved, and an efficient image restoration (red arrows).

One possible solution to narrow the quality gap is to train the StyleGAN generator on low-quality images. Although this might improve GAN inversion accuracy, it can possibly fail to model the high-quality image distribution, which is the desired target of standard image generation and many image manipulation tasks. Training the generator on mixed quality data also does not help since the connection to translate between low-quality and high-quality images is missing.

This paper resolves the aforementioned problems by introducing a novel **Q**uality **C**ontrollable **StyleGAN** structure, or **QC-StyleGAN**, which is a simple yet effective architecture that can learn and represent a wide range of image degradations and encode them in a controllable vector. We demonstrate our solution by modifying StyleGAN2-Ada networks but it should be applicable to any StyleGAN version. Based on the standard structure, we revise its fine-level layers to input a quality code $q$ defining the degradations on the model output. It can generate clean and sharp images similar to the standard StyleGAN counterpart when $q = 0$ and synthesize their degraded, low-quality versions by varying the value $q$. Our QC-StyleGAN covers popular degradations, including noise, blur, low-resolution and downsampling, JPEG compression artifacts, and their mixtures. It has many desired properties. First, QC-StyleGAN can generate sharp and clean images with almost the same quality as the standard StyleGAN, thus preserving all of its utilities and applications. Second, it can also model the degraded photos, bridging the gap to real-world images and providing more accurate GAN inversion. The image editing operators, therefore, can be applied on low-quality images in the same way as for the high-quality ones without altering image quality. This functionality is particularly important when only a part of the data is manipulated, and the manipulated result must have consistent quality as the rest. One scenario is to edit a few frames in a video. Another scenario is to edit an image crop of a big picture. Third, it allows easy conversion between low- and high-quality outputs, bringing in many applications below.

One of our QC-StyleGAN's applications is GAN-based image restoration. Given a low-quality input image, we can fit it into QC-StyleGAN, then recover to the high-quality, sharp version by resetting the quality code $q$ to 0. While there were several works that used the StyleGAN prior for image super-resolution [17, 18] or deblurring [19], QC-StyleGAN is the first method to solve the general image restoration task. Moreover, while the previous methods tried to fit low-quality data to the high-quality image space, leading to obvious content mismatches, our model maintains the content consistency by bridging the two data realms. This image restoration, while being a simple by-product of our QC-StyleGAN design, shows great potential in handling images with complex degradations.

Besides, by modeling a wide range of image degradations and encoding them in a controllable vector, QC-StyleGAN can be used to synthesize novel image degradations or interpolate between existing ones. It can easily capture the degradation from an input image, allowing degradation transfer. These techniques have various practical applications and will be demonstrated in Section 4.

Fig. 1 summarizes our proposed QC-StyleGAN. It covers not only the sharp, high-quality image domain of common StyleGAN models but also the degraded, low-quality image one. It bridges these image domains via a quality control input. QC-StyleGAN allows better GAN inversion and image editing on low-quality inputs and introduces a potential image restoration method.

## 2 Related work

### 2.1 StyleGAN series

Since the seminal paper [20], Generative Adversarial Networks (GANs) have achieved tremendous progress. Among them, a typical line of work called Style-based GANs (StyleGANs) has attracted much attention from the research community. These works allow us to generate images at very high resolution while producing semantically disentangled and smooth latent space. In the initial version [1], StyleGAN controls the style of synthesized images by proposing an intermediate disentangled latent space, named $\mathcal{W}$ space, mapped from the latent code via an MLP network. Then, they feed this code to the generator at each layer by employing the adaptive instance normalization module. In the next generation, StyleGAN2 [2] proposed a few changes in the network design and training components to further enhance the image quality. StyleGAN-Ada [3] allows model training with limited data by introducing the adaptive discriminator augmentation technique. StyleGAN3 [4] tackles the aliasing artifact phenomenon in the previous versions and therefore helps to generate images entirely equivariant for rotation and translation. Recently, StyleGAN-XL [21] expanded the ability of the StyleGAN model to synthesize images on the ImageNet [22] dataset. It is worth noting that all of the current StyleGAN models have used the training dataset with high-quality images. Therefore, their outputs also are sharp images. In our work, we explore a new StyleGAN structure that allows us to synthesize both high- and low-quality images with explicitly quality control.

### 2.2 Latent space traversal and GAN inversion

Aside from the ability to synthesize high-quality images, the latent space learned by GANs also encodes a diverse set of interpretable semantics, making it an excellent tool for image manipulation. As a result, exploring and controlling the latent space of GANs has been the focus of numerous research works. Many studies [23, 24, 25] have tried to extract the editing directions from the latent space in a supervised manner by leveraging either a pre-trained attribute classifier or a set of attribute-annotated images. Meanwhile, other works such as [15, 26, 27, 28] have developed the unsupervised methods for mining the latent space, which reveal many new interesting editing directions. To convey such benefits for editing real images, we first need to obtain the latent code in the latent space so that we can accurately reconstruct the input image when we feed this code into the pre-trained generator. This line of work is called *GAN inversion*, which was first proposed by [29]. Existing GAN inversion techniques can be grouped into (1) optimization-based [30, 31, 7, 2, 32]; (2) encoder-based [29, 33, 9, 10, 34] and (3) two-stage [35, 8, 36, 37, 38, 39, 13, 12, 40] approaches. We recommend visiting the comprehensive survey [41] for a more in-depth review.

### 2.3 Image enhancement and restoration

Image enhancement and restoration are the task of increasing the quality of a given degraded image. Formally, the degradation process can be generalized as:

$$y = \mathcal{H}(x) + \eta \tag{1}$$

where $\mathcal{H}$ is the degradation operator, $\eta$ is noise, $x$ and $y$ are the original and the degraded images, respectively. The goal is to find $x$ given $y$ and probably some assumptions on $\mathcal{H}$ and $\eta$. It can be divided into various sub-tasks based on the degraded operator and the noise such as image denoising ($\mathcal{H} = I$), image deblurring ($\mathcal{H}$ is a blur operator), or image super-resolution ($\mathcal{H}$ is a downsample operator). Existing methods usually focus on one of these sub-tasks instead of solving the general one. Image enhancement and restoration is a well-studied yet still challenging field. In the past, common methods make handcrafted priors on $\mathcal{H}$ and $\eta$ and use complicated optimization algorithms to solve $x$ [42, 43, 44, 45]. Recently, many deep-learning-based methods have been proposed and achieved impressive results. These methods mainly differ by the network design [46, 47, 48]. However, data-driven approaches were observed to be highly overfitted to the training set and hence cannot be applied for real-world degraded images.

To better restore in-the-wild images, recent works consider the task on a specific domain, such as face [17, 18, 49], by leveraging existing generative models, such as StyleGAN. Unlike previous deep-based models, these methods always produce high-quality results even on real-world degraded images. However, they often produce clear mismatched image content when trying to fit the degraded input into the sharp image space.

# 3 Proposed method

In this section, we present our proposed QC-StyleGAN that supports quality-controllable image generation and manipulation. We first define the QC-StyleGAN concept (Section 3.1), then discuss its structure (Section 3.2) and training scheme (Section 3.3). Next, we discuss the technique to acquire precise inversion results (Section 3.4). Finally, we present its various applications (Section 3.5).

## 3.1 Problem definition

Traditional image generators input a random noise vector $z \in \mathbb{R}^{D_i} \sim \mathcal{N}(0, I)$, with $D_i$ as the number of input dimensions, and output the corresponding sharp image. Let us denote the baseline StyleGAN generator as $\mathcal{F}_0$. The image generation process is $I = \mathcal{F}_0(z)$. Furthermore, $\mathcal{F}_0$ consists of two components: a mapping network $\mathcal{M}_0$ and a synthesis network $\mathcal{G}_0$:

$$\mathcal{F}_0 = \mathcal{G}_0 \circ \mathcal{M}_0, \qquad I = \mathcal{F}_0(z) = \mathcal{G}_0(\mathcal{M}_0(z)) = \mathcal{G}_0(w), \tag{2}$$

with $w = \mathcal{M}_0(z)$ forming a commonly used embedding space $\mathcal{W}$.

Our proposed network, denoted as $\mathcal{F}$, requires an extra quality code input, denoted as $q \in \mathbb{R}^{D_q} \sim \mathcal{N}(0, I)$, with $D_q$ as the number of quality-code dimensions. Its image generation process is $I = \mathcal{F}(z, q)$. We borrow the mapping function $\mathcal{M}_0$ and design a new synthesis network $\mathcal{G}$ for $\mathcal{F}$:

$$\mathcal{F} = \mathcal{G} \circ \mathcal{M}_0, \qquad I = \mathcal{F}(z, q) = \mathcal{G}(\mathcal{M}_0(z), q) = \mathcal{G}(w, q). \tag{3}$$

The desired network should satisfy two requirements:

1. When $q$ is zero, $\mathcal{F}$ synthesizes sharp images similar to the standard StyleGAN:

$$\mathcal{F}(z, 0) = \mathcal{F}_0(z). \tag{4}$$

2. When $q$ is nonzero, $\mathcal{F}$ synthesizes a degraded version of the corresponding sharp image:

$$\mathcal{F}(z, q) = \mathcal{A}_q(\mathcal{F}_0(z)), \tag{5}$$

where $\mathcal{A}_q$ is some image degradation function with the parameter $q$. $\mathcal{A}_q$ is a composition of multiple primitive degradation functions such as noise, blur, image compression, and more. It is defined by $q$ and independent to the sharp image content $\mathcal{F}_0(z)$.

## 3.2 Network structure

The structure of QC-StyleGAN is illustrated in Fig. 2a. As mentioned, it consists of two sub-networks: the mapping network $\mathcal{M}_0$ borrowed from the standard StyleGAN and a synthesis one $\mathcal{G}$. We will skip $\mathcal{M}_0$ and focus on the design of $\mathcal{G}$. We build $\mathcal{G}$ from the standard synthesis network $\mathcal{G}_0$ with minimal modifications to keep similar high-quality outputs when $q = 0$. It consists of $N$ synthesis blocks, corresponding to different resolutions from coarse to fine. Since image degradations affect high-level details, we revise only the last $M$ layers of $\mathcal{G}$ to input the quality code $q$ and synthesize the corresponding degradation. We empirically found $M = 2$ enough to cover all common degradations.

For each revising synthesis block, we pick the output of the last convolution layer, often named $conv1$, to revise by adding feature residuals conditioned on the quality input $q$. To do so, we introduce a novel network block, called DegradBlock, and plug it in as illustrated in Fig. 2a. Let us denote the input of $conv1$ as a feature map $f \in \mathbb{R}^{C \times H \times W}$ with $C$ as the number of channels and $(H, W)$ as the spatial resolution. DegradBlock, denoted as a function $DB(\cdot)$, inputs $f$ and $q \in \mathbb{R}^{D_q}$ and outputs a residual $r \in \mathbb{R}^{C' \times H \times W}$ with $C'$ as the number of output channels of $conv1$.

When $q = 0$, we expect $\mathcal{G}$ to behave similar to $\mathcal{G}_0$. Hence, in that case, we enforce the network features unchanged, or there is no residual:

$$DB(f, 0) = 0 \quad \forall f. \tag{6}$$

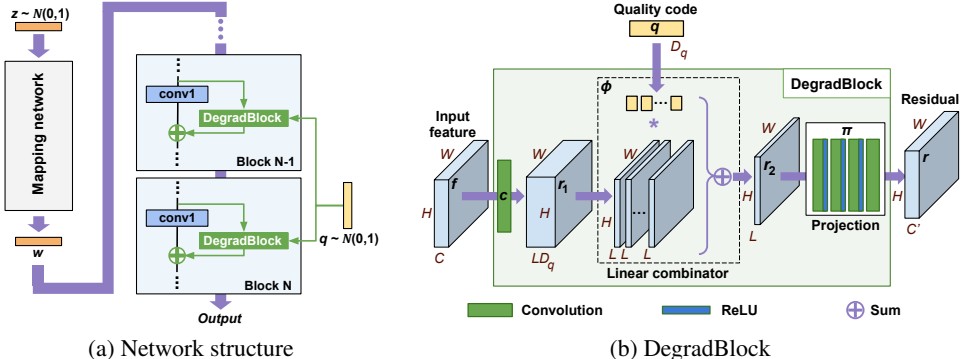

(a) Network structure            (b) DegradBlock

Figure 2: QC-StyleGAN structure

Also, when magnifying $q$, we expect the strength of image degradation, implied by the residual $r$, to increase. From those desired properties, we propose the structure of DegradBlock as in Fig. 2b. It comprises the following components:

1. A convolution layer $c$ with stride 1 to change the number of channels to be a multiple of $D_q$, denoted as $L * D_q$:

$$r_1 = c(f) \in \mathbb{R}^{LD_q \times H \times W}. \tag{7}$$

2. A linear combinator $\phi$ that splits the previous output into $D_q$ $L$-channel tensors $\{r_1^{(i)}\}$, then computes their linear combination using the weights defined by $q$:

$$r_2 = \phi(r_1) = \sum_{i=1}^{D_q} q_i * r_1^{(i)} \in \mathbb{R}^{L \times H \times W}. \tag{8}$$

3. A projection module $\pi$ to refine $r_2$ and change its number of channels to $C'$:

$$r = \pi(r_2) \in \mathbb{R}^{C' \times H \times W}. \tag{9}$$

We design $\pi$ as as a stack of $P$ convolution layers with stride 1. To increase non-linearity, we put a ReLU activation after each convolution layer, except the last one. Note that when $q = 0$, we expect $r = 0$ (Equation 6) and also have $r_2 = 0$ (according to Equation 8). It leads to $\pi(0) = 0$. We ensure it by simply setting the convolution layers to have no bias.

This design is inspired by PCA, unlike the common-used AdaIN blocks. We first use $c$ to predict $D_q$ principal components $r_1^{(i)}$, then compute their linear combination with $q$ as the component weights. This structure is simple but satisfies the mentioned properties. When $q = 0$, the residual is guaranteed to be 0. When magnifying $q$, the residual increases accordingly (see the Appendix):

$$DB(f, k * q) = k * DB(f, q) \quad \forall k \in \mathbb{R}. \tag{10}$$

## 3.3 Network training

Next, we discuss how to train our QC-StyleGAN. As mentioned, it differs from the standard StyleGAN only on the last two blocks of the synthesis network. Hence, we initiate our network from the pretrained StyleGAN weights and finetune only those two synthesis blocks.

Our QC-StyleGAN is trained in two modes corresponding to sharp ($q = 0$) and degraded ($q \neq 0$) image generation. We have a sharp-image discriminator $\mathcal{D}_s$ and a degraded-image discriminator $\mathcal{D}_d$ used in each mode. In the degraded image generation mode, we augment the sharp images by a random combination of primitive degradation functions (noise, blur, image compression) to get "real" low-quality images for training the discriminator.

For each mode, we train the networks with similar losses as in the standard StyleGAN training. However, in the sharp image generation mode, we employ the standard StyleGAN $\mathcal{F}_0$ as the teacher model and apply knowledge distillation to further ensure similar sharp image outputs. We transfer

knowledge in the *feature* spaces instead of the output space for more efficient distillation. Let $S_{KD}$ denotes the set of $conv1$ layers with added DegradBlocks, $X_0^{(l)}$ and $X^{(l)}$ denote features of layer $l^{th}$ in $S_{KD}$ from the teacher and student networks. We define the extra distillation loss $\mathcal{L}_{dist}$ as follows:

$$\mathcal{L}_{dist} = \sum_{l \in S_{KD}} \|X_0^{(l)} - X^{(l)}\|^2. \tag{11}$$

This distillation loss is added to the final loss with a weighting-hyper-parameter $\lambda_{KD}$.

### 3.4  Inversion process

After getting the QC-StyleGAN model, we next discuss how to fit any input image to its space. This process, called GAN Inversion, is a critical step in many applications such as image editing. While the general objective is to reproduce the input image, different methods optimize different components of the image generation process. We follow the state-of-the-art technique named PTI [39] to optimize the $w$ embedding and the generator $\mathcal{G}$ to acquire both precise reconstruction and high editability. Furthermore, we also need to optimize the newly proposed quality-control input $q$. Let us revise the denotation of the synthesis network $\mathcal{G}$ as $\mathcal{G}_\theta$ with $\theta$ as its weights. Our inversion task $\mathcal{I}_\mathcal{G}$ estimates both the inputs $(w, q)$ and lightly tunes $\theta$ so that the reconstructed image is close to the input:

$$\mathcal{I}_\mathcal{G}(I, \theta_0) = (w^*, q^*, \theta^*) = \underset{w,q,\theta}{\operatorname{argmin}} \, d(\mathcal{G}_\theta(w, q), I) \qquad \text{given that } \|\theta - \theta_0\| < \epsilon, \tag{12}$$

where $I$ is the input image, $d(\cdot)$ is a distance function, $\theta_0$ is the network weights acquired from Section 3.3, and $\epsilon$ is some threshold restricting the network weight change.

PTI [39] proposes a two-step inversion process. It first optimizes the embedding $w$ using the initial model weights $\theta_0$ (stage-1), then keeps the optimized embedding and finetunes $\theta$ (stage-2). We can adapt that process to QC-StyleGAN, with a small change to include $q$ in optimization alongside $w$.

However, one extra requirement for this inversion, specific to QC-StyleGAN, is to have the sharp version of the reconstructed image, i.e., $\mathcal{G}_\theta(w^*, 0)$, to be high-quality. We empirically found that the naive optimization processes in PTI fail to achieve that goal. One degraded image may correspond to different sharp images, e.g., in case of motion or low-resolution blur. PTI, while manages to nicely fit the degraded input, often picks non-optimal embeddings that produce distorted corresponding sharp images. Hence, we replace its stage-1 with a training-based approach, following pSp [9], with extra supervision on the sharp image domain. In this **revised stage-1**, we train two encoders to regress the embedding $w$ and the quality-code $q$ separately. Also, we load both the degraded and the corresponding sharp images for training and apply reconstruction losses on both image versions.

### 3.5  Applications

**Image editing**. The most intriguing application of StyleGAN models is to manipulate real-world images with realistic attribute changes. They can do that by applying learned editing directions in some embedding space, e.g., the $\mathcal{W}$ space. However, these standard models can only do the editing in sharp image domains. Our QC-StyleGAN inherits the editing ability of StyleGAN by using the same network weights except for the last $M$ synthesis blocks, which mainly affect the fine output details. However, QC-StyleGAN covers both low- and high-quality images, broadening the application domains. It can directly apply the learned editing directions of StyleGAN on low-quality images and keep their degradations unchanged. Specifically, given an input $I$ and a target editing direction $\Delta w$ learned from the standard StyleGAN in the $\mathcal{W}$ space, we can first invert the image $(w, q, \theta) = \mathcal{I}_\mathcal{G}(I, \theta_0)$, then generate the manipulated result $I' = \mathcal{G}_\theta(w + \Delta w, q)$.

**Image restoration**. This is a new functionality, which is a by-product of QC-StyleGAN design. Given a low-quality image $I$, we can fit it to QC-StyleGAN $(w, q, \theta) = \mathcal{I}_\mathcal{G}(I, \theta_0)$, then acquire its sharp, high-quality version by clearing the quality code: $I' = \mathcal{G}_\theta(w, 0)$.

**Degradation synthesis**. QC-StyleGAN defines the degradation on the output image by a quality code $q$. It allows to revise the image degradation in various ways, such as (1) sampling a novel degradation, (2) transferring from another image, and (3) interpolating a new degradation from two reference ones.

Table 1: **FID scores** of our QC-StyleGAN models, in comparison with the baseline StyleGAN2-Ada (SG2-Ada) [3], on sharp and degraded image generation modes. '*' means a new, separate SG2-Ada model trained on degraded images.

| FID | FFHQ (256×256) | | AFHQ Cat (512×512) | | LSUN Church (256×256) | |
|---|---|---|---|---|---|---|
| | SG2-Ada | Ours | SG2-Ada | Ours | SG2-Ada | Ours |
| Sharp | 3.48 | 3.65 | 3.55 | 3.56 | 3.86 | 3.61 |
| Degraded | 4.38* | 3.23 | 4.70* | 3.91 | 5.16* | 4.58 |

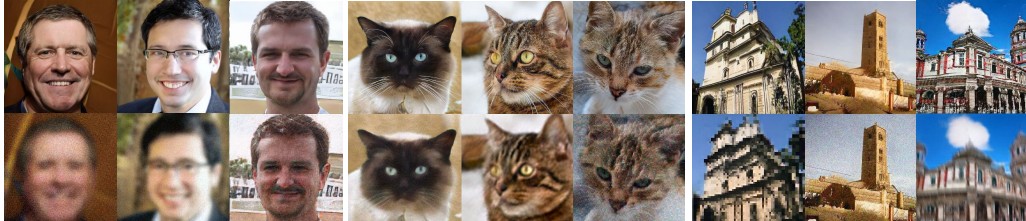

Figure 3: Sample images generated by our models on FFHQ (left), AFHQ-Cat (middle), and LSUN-Church (right). For each sample, we provide a pair of sharp (top) and degraded (bottom) images.

## 4 Experiments

### 4.1 Experimental setup

**Datasets**. We conduct experiments on the common datasets used by StyleGAN, including FFHQ, AFHQ-Cat, and LSUN-Church. FFHQ [1] is a large dataset of 70k high-quality facial images collected from Flickr, introduced since the first StyleGAN paper. We will use the FFHQ images with the resolution $256 \times 256$. AFHQ [50] is a HQ dataset for animal faces with image resolution $512 \times 512$. We demonstrate our method using its Cat subset with about 5000 images. Finally, LSUN-Church is a subset of the LSUN [51] collection. It has about 126k images of complex natural scenes of church buildings at the resolution $256 \times 256$. We will use LSUN-Church only for image generation since its inversion results even on sharp image domain are not satisfactory [13].

**Synthesis network**. We use StyleGAN2-Ada as reference to implement our QC-StyleGAN. The quality code has size $D_q = 16$. In DegradBlock, we use $L = 32$ and $P = 3$. The weight for the distillation loss $\lambda_{KD} = 3$. Our networks were trained using the same settings as in the original work until converged. Details of this training process will be provided in the Appendix.

### 4.2 Image generation

We compare the quality of our QC-StyleGAN models with their StyleGAN2-Ada counterparts in Table 1, using the FID metric. As can be seen, our models have equivalent results to the baselines when generating sharp images. However, while the common StyleGAN2-Ada models cannot produce degraded images, ours can generate such images directly with good FID scores. Even when training new StyleGAN2-Ada models dedicated on degraded images, their FID scores are worse than ours.

Fig. 3 provides some samples synthesized by our networks. For each data sample, we synthesize a degraded image and the corresponding sharp version. As can be seen, the sharp images look realistic, matching the standard StyleGAN's quality. The degraded images match the sharp ones in content, and the degradations are diverse, covering noise, blur, compression artifacts, and their mixtures.

### 4.3 GAN inversion and Image editing

We now turn to evaluate the effectiveness of our proposed GAN inversion technique (Section 3.4) and image editing (Section 3.5) on low-quality image inputs.

With the model trained on the FFHQ dataset, we use the CelebA-HQ [52, 53] test set for evaluation. With the models trained on AFHQ-Cat, we employ its corresponding test set for testing. For each test set, we apply different image degradations to the original images to obtain the low-quality

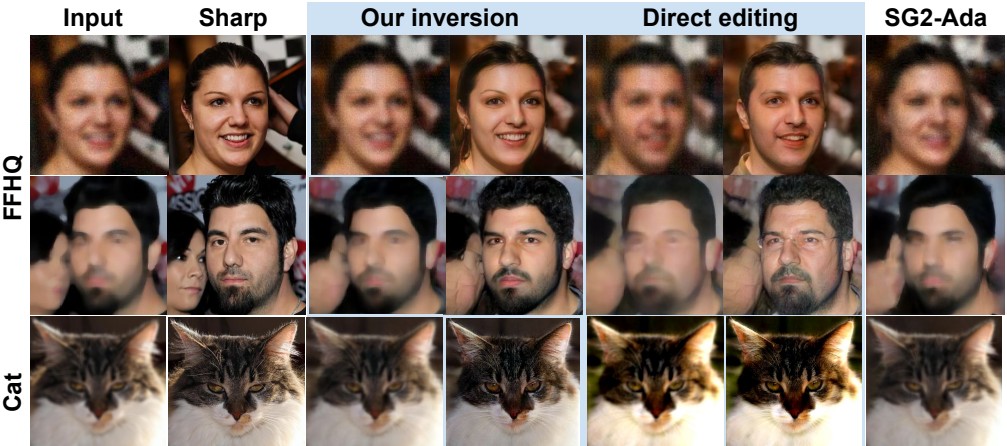

| Input | Sharp | Our inversion | Direct editing | SG2-Ada |
|-------|-------|---------------|----------------|---------|

Figure 4: **GAN Inversion and Image editing.** From a degraded input image ($1^{st}$ col.), we can fit it into our model to get a similar reconstructed degraded image ($3^{rd}$ col.), which is more accurate than from StyleGAN2-Ada (last col.). The corresponding sharp image ($4^{th}$ col.) is close to the real one ($2^{nd}$ col.). We can apply image editing directly on the degraded image ($5^{th}$ col.) and its sharp version ($6^{th}$ col.) matches the change. From top to bottom we apply gender, age, and color change.

images. Our QC-StyleGAN model can fit well to such degraded inputs with the average PSNR of reconstructed images as 29.47dB and 28.91dB for CelebA-HQ and AFHQ Cat, respectively. We also try to apply the image editing directions learned for StyleGAN2-Ada by InterfaceGAN [14] (face) and SeFa [28] (cat) to manipulate the degraded images with QC-StyleGAN. The qualitative results in Fig. 4 confirm the effectiveness of such a direct image editing scheme. Note that while we can also do GAN inversion with the StyleGAN2-Ada model on these inputs, its inversions cannot be converted to sharp. Also, direct manipulation on StyleGAN2-Ada's inversions may introduce unrealistic artifacts (see the Appendix), making StyleGAN inferior to QC-StyleGAN in handling low-quality inputs.

## 4.4 Image restoration

As mentioned in Sec. 3.5, a nice by-product of our QC-StyleGAN is a simple but effective image restoration technique. We examine it on the degraded CelebA-HQ images on five common restoration tracks, including deblurring, super-resolution, denoising, JPEG removal, and multiple-degraded restoration. We also compare it with the state-of-the-art image restoration methods. For image restoration networks such as NAFNet [54] and MPRNet [55], we re-train the models on our degraded images using their published code with default configuration. For GAN-based methods like HiFaceGAN [56] and PULSE [17], we use their provided pre-trained models. The results are reported in Table 2.

Among the common metrics for this task, we find LPIPS more reliable and close to human perception. Although our method is not tailored to handle this restoration task specifically, it performs reasonably well and outperforms many baseline methods in each task. Particularly, QC-StyleGAN provides the best LPIPS score when having multiple degradations in the input. Also, we find that one degraded image may correspond to multiple possible sharp images. Our restoration results sometimes look reasonable but do not match the ground-truth, severely hurting our LPIPS scores.

To avoid the mismatching ground-truth issue, we also use NIQE [57], which is a no-reference image quality metric. QC-StyleGAN provides the best NIQE score in nearly all tracks. It confirms that our image restoration can produce the highest output quality in terms of naturalness [57] while still maintaining comparable perceptual similarity [58] compared to the competitors.

Fig. 5 provides restoration results from our method and the image restoration baselines on two extremely degraded images. Our algorithm manages to return sharp and detailed images, while the others fail to handle such images and show clear artifacts on their recovered images.

| | Track | | | | |
|---|---|---|---|---|---|
| | Blur | Super-res. | Noise | JPEG comp. | Multiple-deg. |
| HiFaceGAN [56] | 5.95 / 0.216 | **5.32** / 0.125 | 6.01 / 0.126 | 4.95 / **0.053** | 5.916 / 0.364 |
| ESRGAN [59] | - | 6.35 / 0.148 | - | - | - |
| DnCNN [47] | - | - | 6.93 / **0.080** | - | - |
| MPRNet [55] | 8.12 / **0.194** | 6.73 / 0.230 | 7.33 / 0.143 | 7.64 / 0.128 | 8.97 / 0.299 |
| PULSE [17] | - | 6.29 / 0.296 | - | - | - |
| mGANPrior [60] | - | 6.02 / 0.265 | - | - | - |
| GLEAN [18] | - | 7.29 / **0.072** | - | - | - |
| Ours | **5.83** / 0.195 | 5.45 / 0.177 | **5.41** / 0.183 | **4.51** / 0.118 | **5.64** / **0.260** |

Table 2: NIQE [57] and LPIPS [58] scores of image restoration methods on five restoration tracks on the CelebA-HQ dataset. For both metrics, lower value means better. The **best** and runner-up values are marked in bold and underline, respectively. The mark '-' means the method is not applicable.

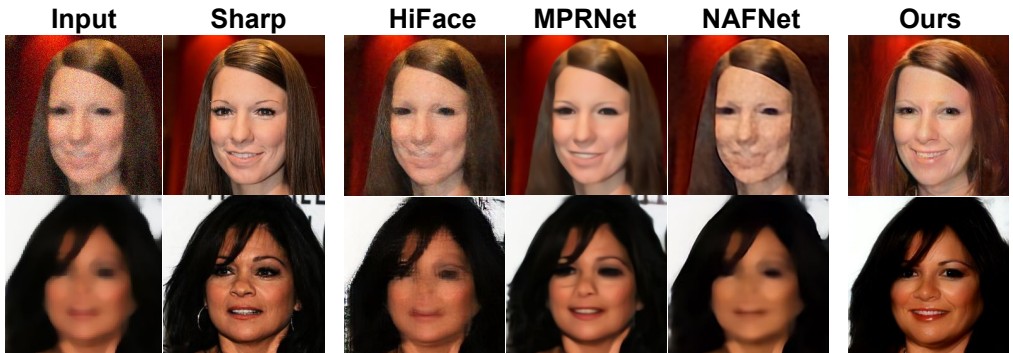

Figure 5: Comparison between our method and image restoration baselines on CelebA-HQ dataset.

## 4.5 Degradation synthesis

We provide an example of our proposed image degradation synthesis (Section 3.5) in Fig. 6. From a source image with JPEG compression artifacts, we change its image degradation to a novel random one (blur, $2^{nd}$ col.) or copy the degradation from a reference image (noise, $6^{th}$ col.). We can also smoothly interpolate in-between degradations, using an interpolation factor $\alpha \in [0, 1]$ ($3 - 5^{th}$ col.).

## 4.6 Quality-code-based Degradation Classification

To verify the quality of QC-StyleGAN in degradation estimation, we conduct experiments of training linear classifiers to predict whether an image is blurry, or at which blur level, based on its quality code $q$. For each experiment, we use 1000 facial images for training and 200 images for testing. When we train the classifier to detect if an image is blurry, the accuracy is **97.9%**. When we divide the degree of blurring into 5 levels for the classifier to predict, the accuracy is 85%. When we use 10 blur levels, the accuracy is still high (77%), confirming QC-StyleGAN as a quality degradation estimator.

## 4.7 Stability of Inversion and Editing

We conduct experiments to verify the stability of our inversion and editing on degraded CelebA-HQ inputs images. We tried the editing magnitudes $\pm 3$ for 3 editing tasks on gender, age, and smiling. For each input image and each editing operator, we executed the operator 3 times to get 3 manipulated outputs and compare them pairwise using the PSNR and LPIPS metrics. The manipulated degraded images are pretty similar, with the PSNR score $44.42 \pm 2.71$ and the LPIPS score $0.004 \pm 0.003$. When recovering the sharp version of these images, the PSNR score is still high ($41.84 \pm 2.07$) and the LPIPS score is still very good ($0.006 \pm 0.005$).

We generated some quantitative videos and provided them in this link. For each video, we show the same manipulation in 6 runs. As can be seen, our manipulated results are quite stable, with minor flickers mainly appearing on the background or the hair region. If we mask out the background

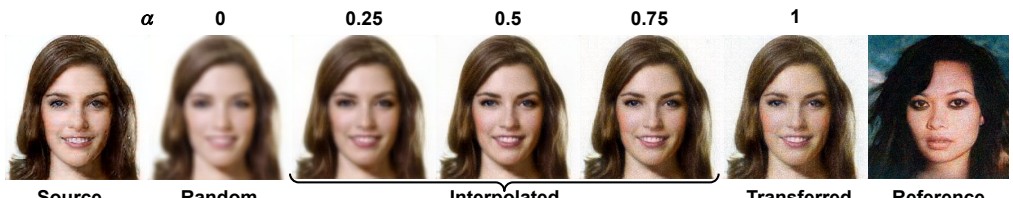

Figure 6: Degradation synthesis

and keep only the face region (the hair region is still kept), the scores in the previous experiments get improved: For degraded images, the PSNR score is $47.18 \pm 2.27$, and the LPIPS score is $0.0013 \pm 0.0006$. For sharp-recovered images, the PSNR score is $43.67 \pm 2.28$, and the LPIPS score is $0.0028 \pm 0.0013$. It confirms that our inversion and editing are pretty stable on the target object.

## 4.8 Ablation studies

In this section, we investigate the effectiveness of our model design on the FFHQ dataset. First, we try other ways to inject $q$ into the network, e.g., concatenating $q$ with the latent input $z$, the embedding $w$, or via DualStyleGAN structure [61], but find clear mismatches in image content of generated sharp and degraded images of the same code $w$. Second, we try our network training without the distillation loss, and the FID-sharp is very high at $11.32$. Third, we try a AdaIN-style design for the DegradBlocks, and its FID-sharp is $4.41$. If we apply DegradBlock at the last synthesis block ($M = 1$), the FID-sharp is $6.38$. If we reduce $D_q$ to $8$, that FID-sharp increases ($4.54$) even if we expand $L$ to $64$ for a similar computation cost. Extra discussions will be included in the Appendix.

## 4.9 Inference time

In this section, we report the running time of the proposed models in Table 3. For the restoration tasks, we report the running time of the two components of the method, including pSp (stage-1) and PTI optimization (stage-2).

Table 3: **Average running times** of our methods on two tasks, including image generation and image inversion (pSp and PTI optimization). Note that the performance of pSp is the same on both resolutions because it resizes the input to $256 \times 256$ before running.

| Model | Time (s) | |
| --- | --- | --- |
| | $256 \times 256$ | $512 \times 512$ |
| QC-StyleGAN | $0.08 \pm 0.01$ | $0.20 \pm 0.04$ |
| pSp | $0.13 \pm 0.01$ | $0.11 \pm 0.01$ |
| PTI-opt | $83.40 \pm 2.68$ | $222.29 \pm 23.24$ |

## 5 Conclusions and future work

This paper presents QC-StyleGAN, a novel image generation structure with quality-controlled output. It inherits the capabilities of the standard StyleGAN but extends to cover both high- and low-quality image domains. QC-StyleGAN allows direct manipulation on in-the-wild, low-quality inputs without quality changes. It offers novel functionalities, including image restoration and degradation synthesis.

**Limitations.** Although we employed many image degradations in training QC-StyleGAN, they might not cover all in-the-wild degradations. Also, we only implemented QC-StyleGAN from StyleGAN2-Ada. We plan to improve on these aspects in the future.

**Potential negative societal impacts.** Our work may have negative societal impacts by allowing better image manipulation on in-the-wild data. However, image forensics is an active research domain to neutralize that threat. We believe our work's benefits dominate its potential risks.

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
