# QC-StyleGAN - Quality Controllable Image Generation and Manipulation – Supplementary Material –

**Dat Viet Thanh Nguyen**[1,*]    **Phong Tran**[1,2,*]    **Tan M. Dinh**[1]

**Anh Tuan Tran**[1]    **Cuong Pham**[1,3]

[1]VinAI Research    [2]MBZUAI    [3]Posts & Telecommunications Institute of Technology

{v.datnvt2, v.tandm3, v.anhtt152, v.cuongpv11}@vinai.io    the.tran@mbzuai.ac.ae

In this supplementary material, we prove theoretical properties and provide additional experiments as well as details of training process and network implementation.

## A  Proofs

### A.1  Proof of Equation 10

First, let us consider the linear combinator $\phi$. When we scale the input $q$, its value is scaled accordingly:

$$\phi(r_1, k*q) = \sum_{i=0}^{D_q} (k*q_i)*r_1^{(i)} = k*\sum_{i=0}^{D_q} q_i*r_1^{(i)} = k*\phi(r_1, q) \quad \forall k \in \mathbb{R}. \tag{1}$$

Next, the projector $\pi$ is a composition of $P' = 2P - 1$ layers:

$$\pi = \pi^{(P')} \circ \pi^{(P'-1)} \circ ... \circ \pi^{(1)} \tag{2}$$

where each layer $\pi^{(i)}$ is either (1) a convolution layer with stride 1 and no bias $\pi_{conv}^{(i)}(x) = W^{(i)}*x$, with $W^{(i)}$ is a weighting matrix, or (2) a RELU layer $\pi_{RELU}^{(i)}(x) = max(x, 0)$. Both of layer types are homogeneous functions with degree 1 since:

$$\pi_{conv}^{(i)}(k*x) = W^{(i)}*(k*x) = k*(W^{(i)}*x) = k*\pi_{conv}^{(i)}(k) \quad \forall k \in \mathbb{R},$$
$$\pi_{RELU}^{(i)}(k*x) = max(k*x, 0) = k*max(x, 0) = k*\pi_{RELU}^{(i)}(x) \quad \forall k \in \mathbb{R}. \tag{3}$$

Hence:

$$\begin{aligned}
\pi(k*x) &= \pi^{(P')}(\pi^{(P'-1)}(...(\pi^{(1)}(k*x)))) \\
&= \pi^{(P')}(\pi^{(P'-1)}(...(k*\pi^{(1)}(x)))) \\
&= ... \\
&= \pi^{(P')}(k*\pi^{(P'-1)}(...(\pi^{(1)}(x)))) \\
&= k*\pi^{(P)}(\pi^{(P'-1)}(...(\pi^{(1)}(x)))) \\
&= k*\pi(x) \qquad\qquad\qquad \forall k \in \mathbb{R}.
\end{aligned} \tag{4}$$

That means the projector $\pi$ is also a homogeneous function with degree 1.

---

[*]authors contributed equally

36th Conference on Neural Information Processing Systems (NeurIPS 2022).

Table 1: Hyperparameters used in each model training.

| Parameter | FFHQ | AFHQ-Cat | LSUN-Church |
|---|---|---|---|
| Resolution | 256×256 | 512×512 | 256×256 |
| Number of GPUs | 8 | 8 | 8 |
| Training length | 5M | 5M | 5M |
| Minibatch size | 64 | 64 | 64 |
| Minibatch stddev | 8 | 8 | 8 |
| Feature maps | $\frac{1}{2}\times$ | $1\times$ | $1\times$ |
| Learning rate $\eta \times 10^3$ | 2.5 | 2.5 | 2.5 |
| $R_1$ regularization $\gamma$ | 1 | 0.5 | 100 |
| Mixed-precision | ✓ | ✓ | ✓ |
| Mapping net depth | 8 | 8 | 8 |
| Style mixing reg. | ✓ | ✓ | ✓ |
| Path length reg. | ✓ | ✓ | ✓ |
| Resnet D | ✓ | ✓ | ✓ |
| Training time | 1.5 days | 3.5 days | 1.5 days |

Finally, we consider the entire DegradBlock:

$$
\begin{aligned}
DB(f, k*q) = \pi(\phi(r_1, k*q)) &= \pi(k*\phi(r_1, q)) && \text{(from Equation 1)} \\
&= k*\pi(\phi(r_1, q)) && \text{(from Equation 4)} \\
&= k*DB(f, q) \quad \forall k \in \mathbb{R}.
\end{aligned}
$$

Thus, the lemma in Equation 10 holds.

## B  Training details

**Hyperparameters**  We build upon the official Pytorch implementation of StyleGAN2-Ada by Karras et al., from which we inherit most of the training details, including weight demodulation, path length regularization, lazy regularization, style mixing regularization, equalized learning rate for all trainable parameters, the exponential moving average of generator weights, non-saturating logistic loss with $R_1$ regularization, and more. We use Adam optimizer with $\beta_1 = 0$, $\beta_2 = 0.99$, and $\epsilon = 10^{-8}$. The quality code has size $D_q = 16$. In DegradBlock, we use $L = 32$ and $P = 3$. The weight for the distillation loss $\lambda_{KD} = 3$. We also report other details for each training in Table 1.

For the pSp model, we use the github project with some modifications to change the generative model to our QC-StyleGAN and add an additional encoder for quality code inversion. Besides, all the network architecture and hyperparameters are kept intact.

**Training environment**  We ran our QC-StyleGAN training on a DGX SuperPOD node with 8 Tesla A100 GPUs, using Pytorch 1.7.1 (for comparison methods), CUDA 11.1, and cuDNN 8.0.5. We use the official pre-trained Inception network to compute FID.

For the image restoration task, we train pSp and run PTI on a single Nvidia V100. It took about 3 days for the pSp training to converge.

**Datasets**  We train our QC-StyleGAN on common datasets, including FFHQ, LSUN-Church, and AFHQ-Cat. For the image restoration task, we train pSp on FFHQ and AFHQ-cat datasets and test the trained models on CelebA and AFHQ-Cat validation sets, respectively. Details of the used datasets are listed in Table 2.

## C  Additional Experimental Results

### C.1  Comparison between Inversion Results of StyleGAN and QC-StyleGAN

The differences between our inversion results and StyleGAN2-Ada inversion ones:

Table 2: **Dataset details.** Asterisks(*) indicate that the set is randomly sampled from the original dataset.

| Dataset | Resolution | #training images | #testing images |
|---|---|---|---|
| FFHQ | $256 \times 256$ | 70000 | - |
| CelebA* | $256 \times 256$ | - | 300 |
| AFHQ-Cat | $512 \times 512$ | 5153 | 300 |
| LSUN Church | $256 \times 256$ | 126227 | - |

- First, our reconstructed images can easily be converted to their sharp version. In contrast, inversion with StyleGAN2-Ada only gives us fitted degraded images.

- Moreover, since QC-StyleGAN models both sharp and degraded images, the inversion results often stay in-distribution, allowing good editing results. In contrast, the original StyleGAN2-Ada network only models sharp images; editing its inversion on degraded inputs might lead to unrealistic outcomes. We provide the comparison between their manipulation results in Fig. 1.

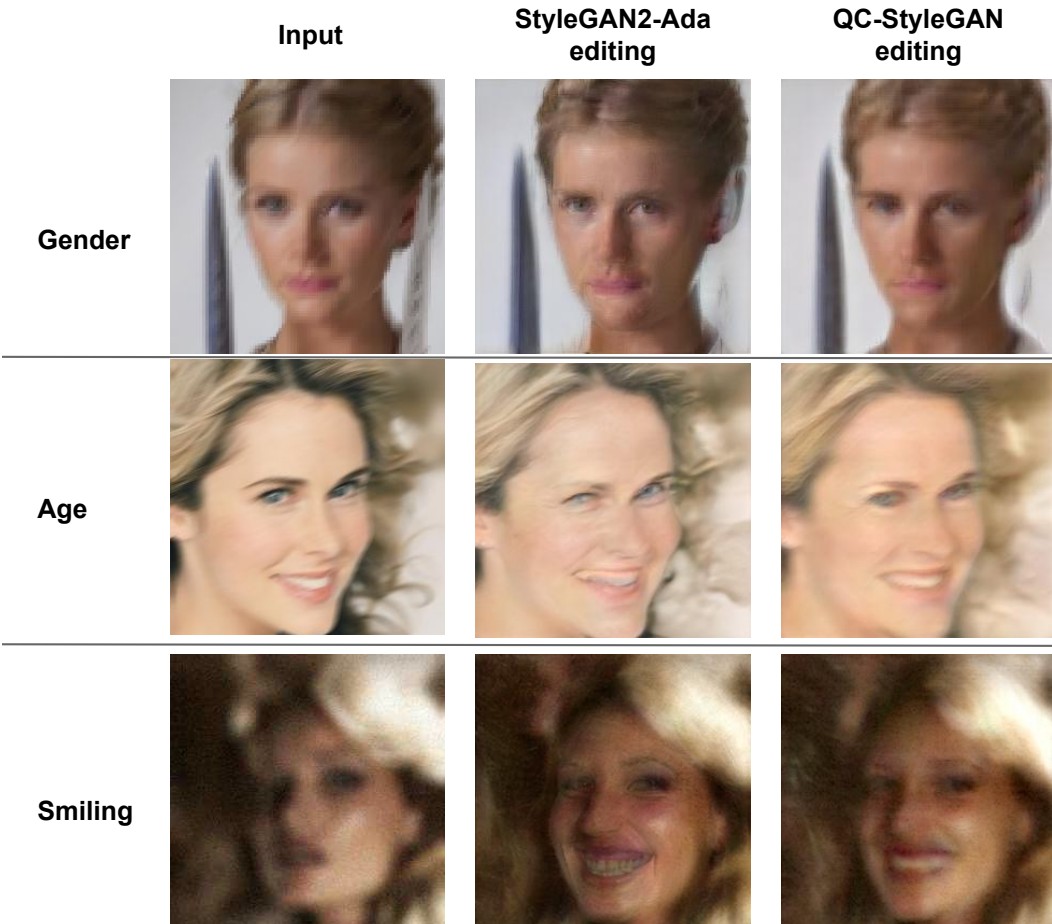

Figure 1: **Image editing comparison.** We compare image editing results based on StyleGAN2-Ada and QC-StyleGAN inversion from degraded FFHQ input images. QC-StyleGAN produces natural-looking images, while StyleGAN2-Ada sometimes produces unrealistic artifacts.

| $\gamma$ | Our (degraded) | Ours (sharp) |
|------|------|------|
| -3.0 | 4.22 | 15.04 |
| -2.33 | 3.39 | 11.79 |
| -1.67 | 2.72 | 9.16 |
| -1.0 | 1.86 | 6.56 |
| -0.33 | 1.12 | 3.5 |
| 0.33 | 0.6 | 0.72 |
| 1.0 | -0.25 | -1.78 |
| 1.67 | -1.19 | -4.24 |
| 2.33 | -1.97 | -6.12 |
| 3.0 | -2.5 | -7.63 |

Table 3: **Quantitative evaluation of editability.** We apply age editing on degraded CelebA-HQ images using different magnitudes $\gamma$ and measure the amount of age change. We test both the manipulated degraded outputs and their sharp recovered version.

## C.2 Quantitative Results on Image Editing

We quantitatively evaluate the editability of our proposed GAN inversion method with QC-StyleGAN by measuring the amount of target attribute changes with different editing magnitudes. Given a degraded image, we synthesize its corresponding edited versions:

$$I_t = G(w + \gamma_t * d, q) \tag{5}$$

where $\gamma_t$ is one of the 10 edit magnitudes uniformly picked in the range of $[-3, 3]$, $w$ and $q$ are the content and quality codes inverted by our proposed inversion method, $d$ is the editing direction explored from the latent space of QC-StyleGAN by using InterfaceGAN [5]. In this test, we choose the age editing direction. To measure the age change when editing, we leverage the off-the-shelf DEX VGG [4] model to estimate the face's age in the images. We perform this experiment on 100 multiple-degraded images and show the average age changes for each magnitude in Table 3. We report results for both the manipulated degraded outputs and their sharp recovered version. The age change scales consistently with the editing magnitude, confirming the editability of the latent space of our generator. Note that the age regression model was trained on sharp images; hence, it produces less significant changes on degraded images, which may not reflect the actual shift on these pictures.

## C.3 Ablation Studies

We report detailed results of our ablation studies in Table 4.

When there is no distillation loss, the training process is unstable and diverges, causing very high FID scores either when finetuning only the last $M$ synthesis blocks (**B**) or the entire network (**A**). It confirms the importance of such distillation loss in our network training.

Next, we investigate the number of synthesis blocks to finetune $M$. When we finetune only 1 block (**C**), the FID score for sharp image generation is 6.38 and the one for degraded image generation is 7.59, which are still high. When $M = 2$ (the official implementation), the FIDs are small, and FID-sharp is close to the one from the standard StyleGAN2-Ada. Since the network performance is already satisfactory, we skip testing with $M > 2$, which requires much more computational cost.

Next, we investigate the design of DegradBlock. We find that increasing the number of layers inside the projection module does not help (**D**), and the FIDs slightly increase.

Next, we examine the choice of the quality code size $D_q$. In our implementation, $D_q$ is set as 16. We tried with a small quality code size with $D_q = 8$, but the FIDs are not as good, even when we increase the number of intermediate channels $L$ to 64 (**E**). It confirms that $D_q$ should be large to cover a wide range of image degradations.

Finally, we try an AdaIN-like DegradBlock (**F**) by replacing the convolution layer $c$ and the linear combinator $\phi$ with an AdaIN module. The FID scores for clean and degraded images on the FFHQ dataset are $4.41$ and $5.28$, respectively, which are significantly higher than our PCA-based DegradBlock design.

Table 4: Ablation studies on our QC-StyleGAN network design.

| Configuration | FID | |
|---|---|---|
| | Sharp | Degraded |
| **A** no distillation loss + no freezed | 11.32 | 26.91 |
| **B** no distillation loss + freezed | 33.45 | 82.11 |
| **C** $M = 1$ layer of $G$ | 6.38 | 7.59 |
| **D** DegradBlock: stack of $P = 5$ convolution layers | 4.02 | 4.68 |
| **E** The quality code has size $D_q = 8$ and $L = 64$ | 4.54 | 5.69 |
| **F** AdaIN-style DegradBlock | 4.41 | 5.28 |

## D  Additional Qualitative Results

### D.1  Image generation

We provide extra image generation results from our QC-StyleGAN on the FFHQ (Fig. 2 and 3), AFHQ-Cat (Fig. 4 and 5), and LSUN-Church (Fig. 6 and 7).

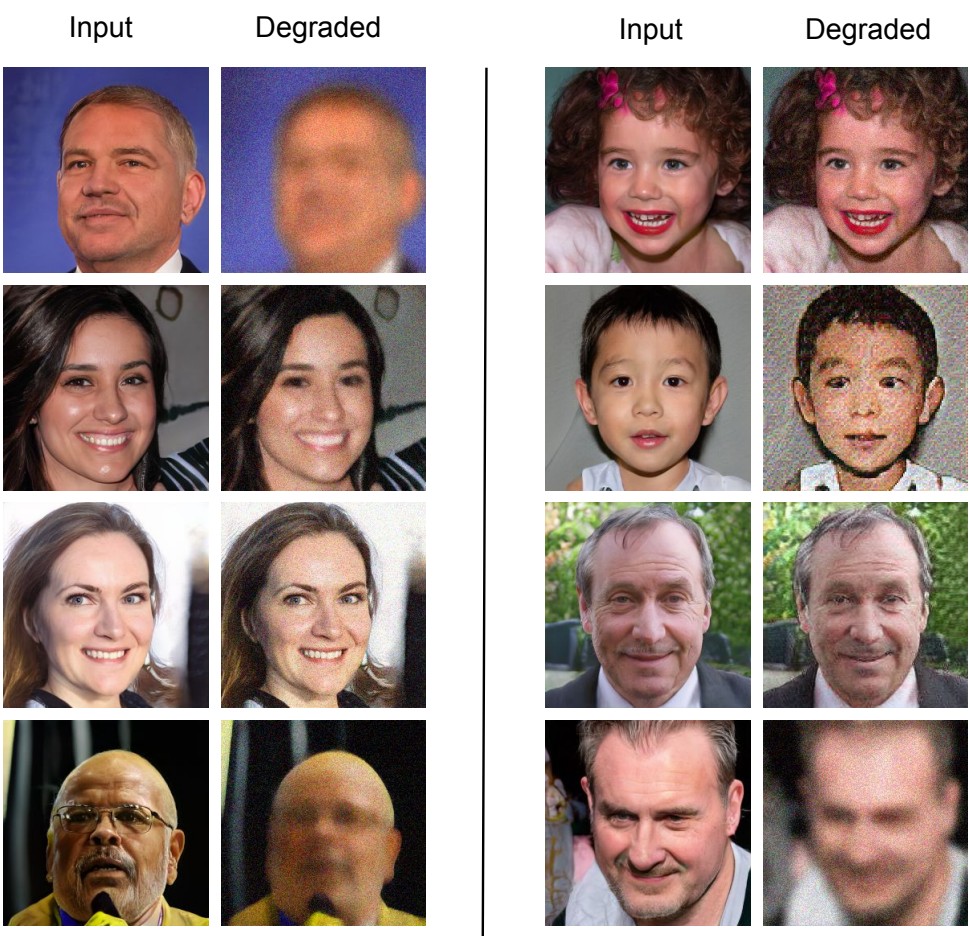

Figure 2: Sample images generated by our QC-StyleGAN trained on the FFHQ dataset.

| Input | Degraded | | Input | Degraded |

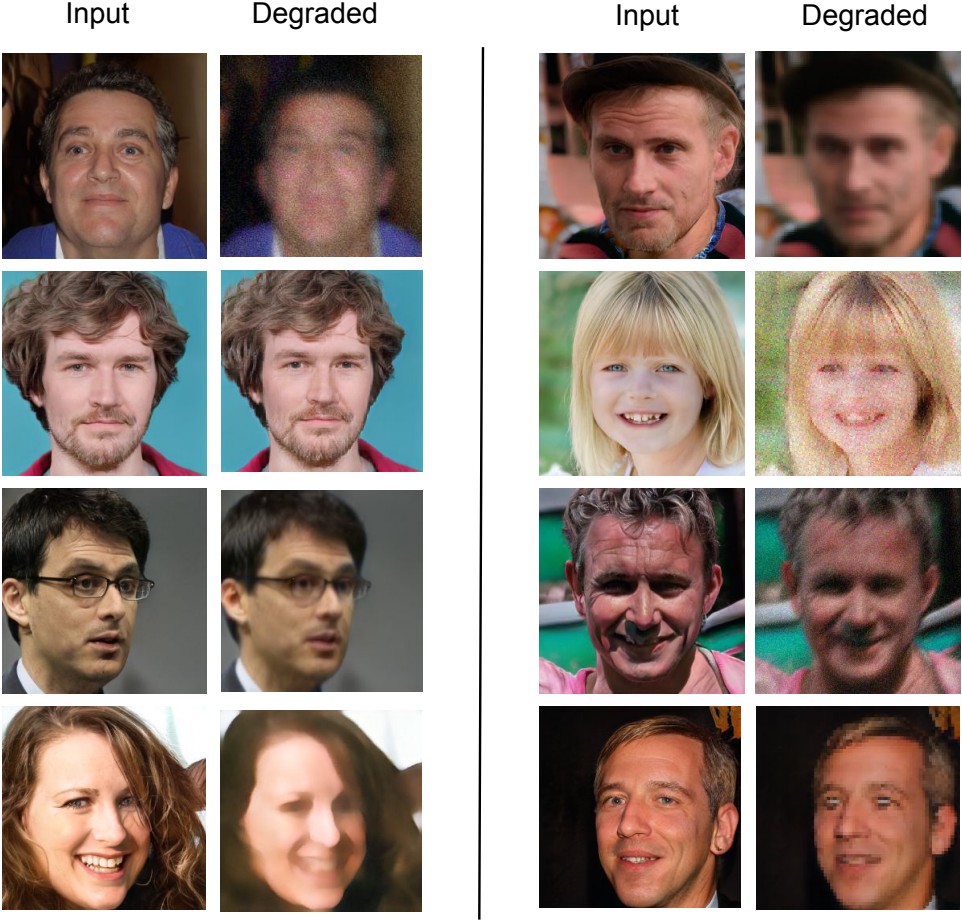

Figure 3: Sample images generated by our QC-StyleGAN trained on the FFHQ dataset.

## D.2 Face image restoration

In this section we provide additional qualitative results of the proposed face image restoration method, compared with state-of-the-art methods. We test two scenarios: image super-resolution and general image restoration.

With the image super-resolution task, each sharp image is downsampled 4 times, using bilinear interpolation, to get the low-resolution input images. The state-of-the-art baselines include PULSE [3], HiFaceGAN [7], Real-ESRGAN [6], NAFNet [1], and MPRNet [8]. Qualitative results are illustrated in Fig. 8, 9, 10, 11, 12, and 13.

With general image super-resolution, the state-of-the-art baselines include HiFaceGAN [7], NAFNet [1], and MPRNet [8]. Qualitative results are illustrated in Fig. 14, 15, 16, 17, 18, 19, 20, 21, 22, 23, 24, and 25.

## D.3 Face editing

In this section, we provide extra image editing results on facial images. We use InterfaceGAN [5] to find the editing direction for "smiling" and "aging". Given a degraded input image, we first perform image inversion, then apply the learned editing on the optimized latent code to change the expression and age of the input face. Qualitative results are given in Fig. 26 and 27.

## D.4 Degradation synthesis

We provide extra image degradation synthesis results in Fig. 28.

Sharp    Degraded              Sharp    Degraded

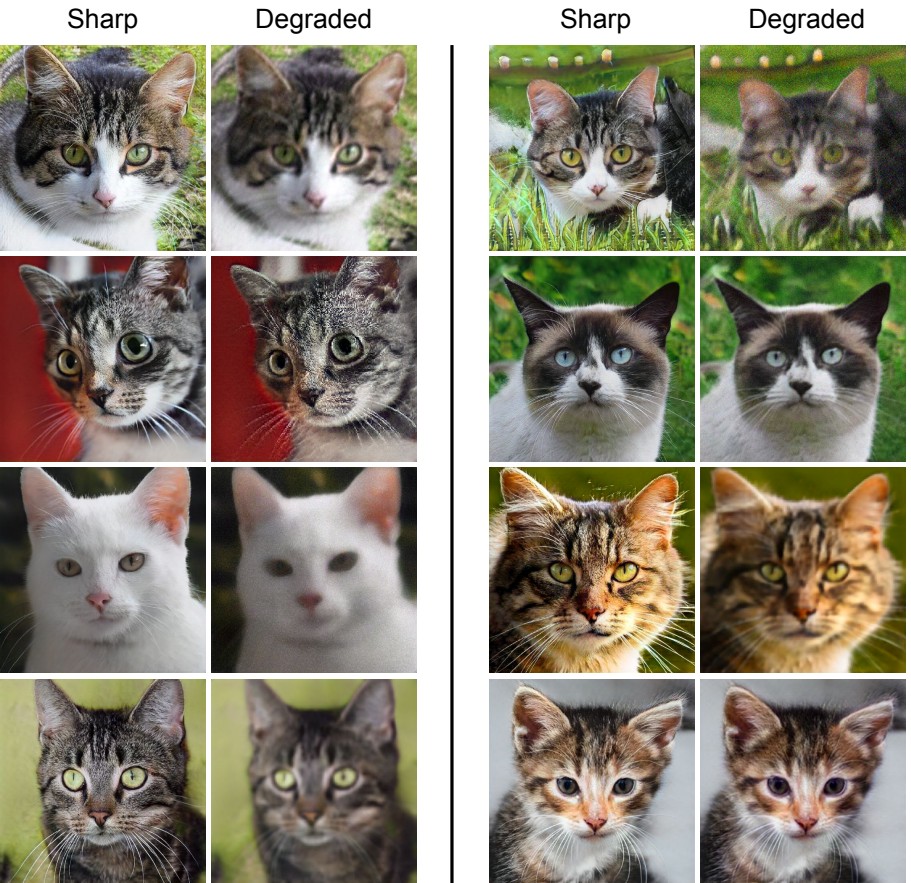

Figure 4: Sample images generated by our QC-StyleGAN trained on the AFHQ-Cat dataset.

### D.5 Image Restoration with Small Degradations

Our QC-StyleGAN models were trained to handle image degradations at various degrees, unlike many deep-learning-based image restoration techniques. We provide some image restoration results on CelebA-HQ images under small degradations, using our QC-StyleGAN model trained on the FFHQ dataset, in Fig. 29. As can be seen, these images are still recovered effectively.

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

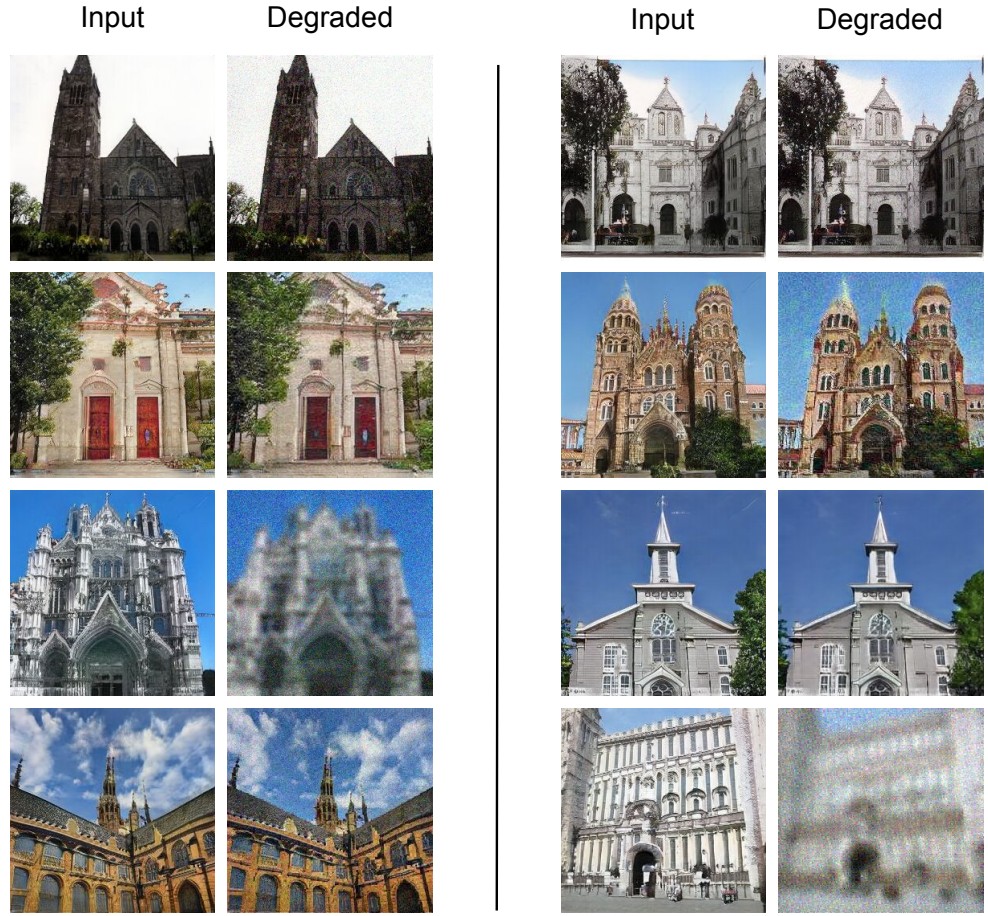

Figure 6: Sample images generated by our QC-StyleGAN trained on the LSUN-Church dataset.

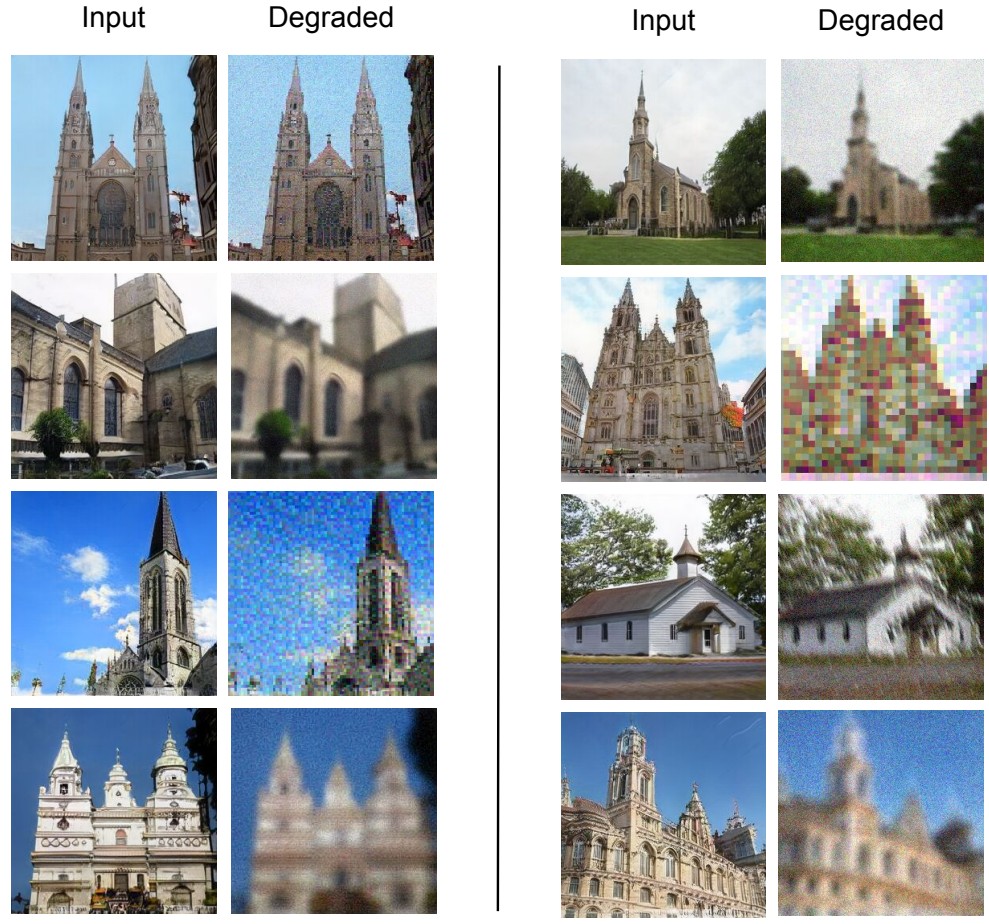

Figure 7: Sample images generated by our QC-StyleGAN trained on the LSUN-Church dataset.

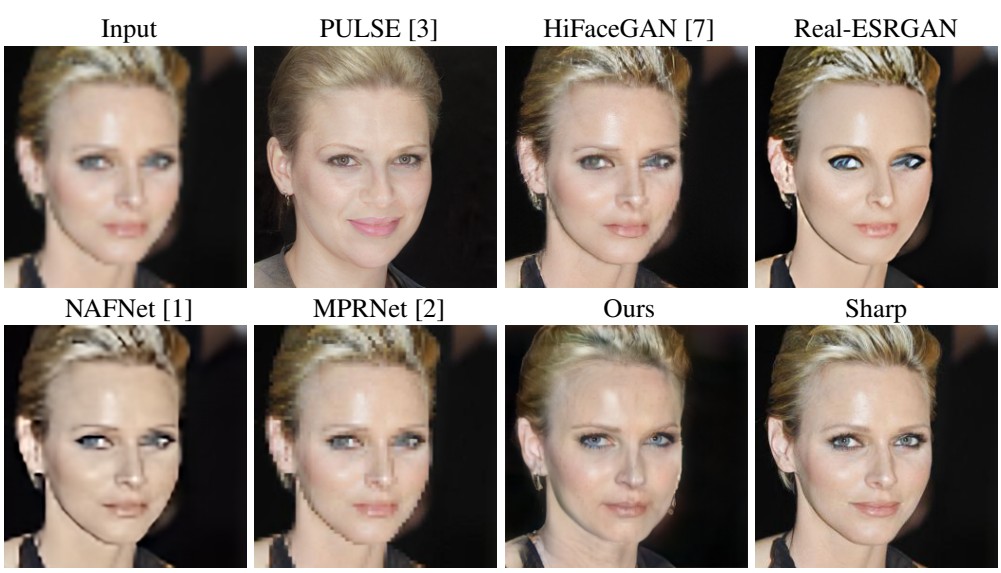

Figure 8: Additional super-resolution qualitative result on the CelebA-HQ dataset

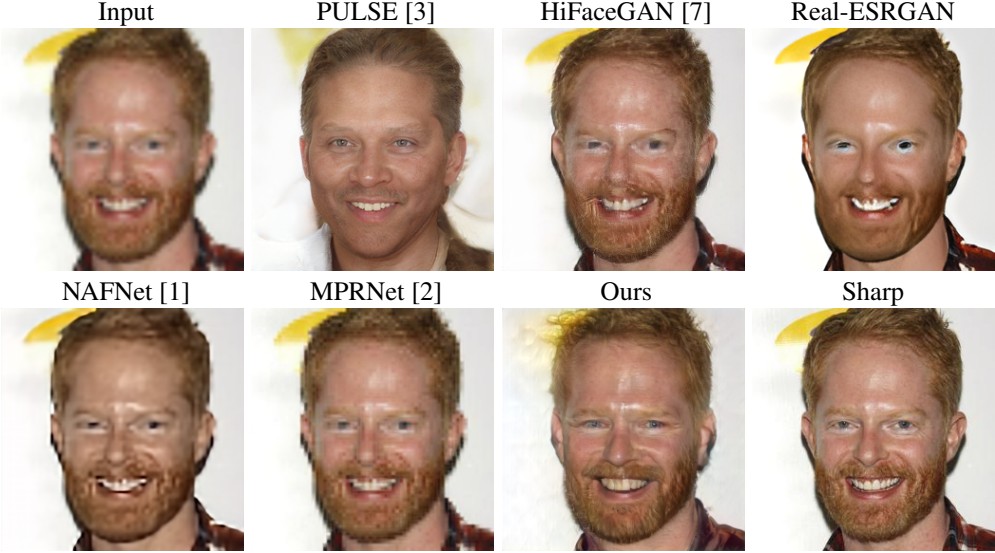

Figure 9: Additional super-resolution qualitative result on the CelebA-HQ dataset

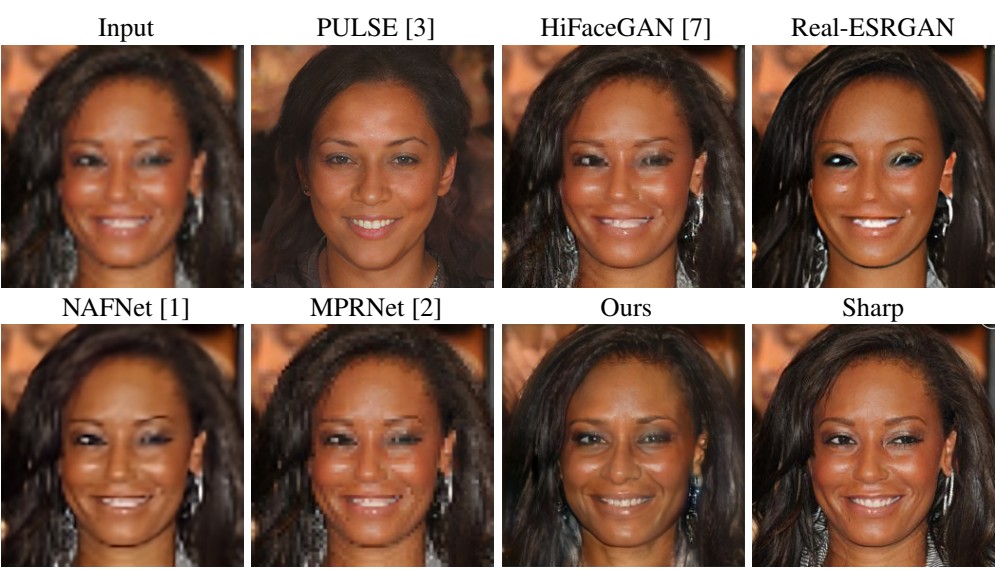

Figure 10: Additional super-resolution qualitative result on the CelebA-HQ dataset

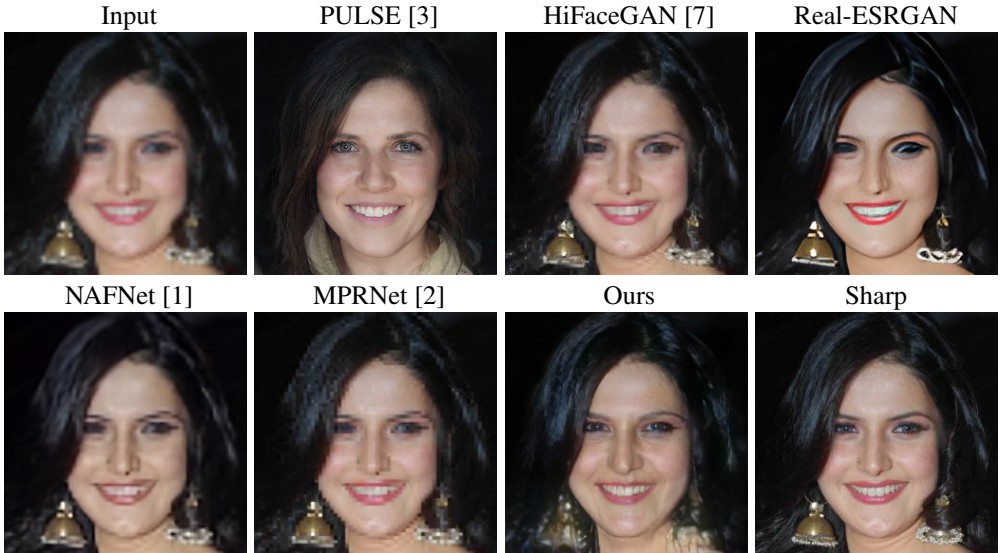

Figure 11: Additional super-resolution qualitative result on the CelebA-HQ dataset

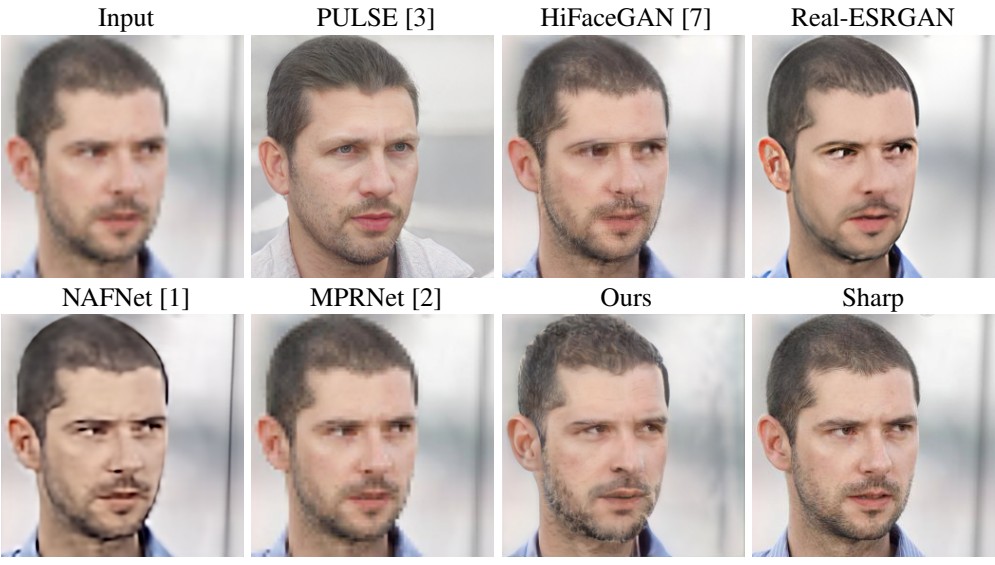

Figure 12: Additional super-resolution qualitative result on the CelebA-HQ dataset

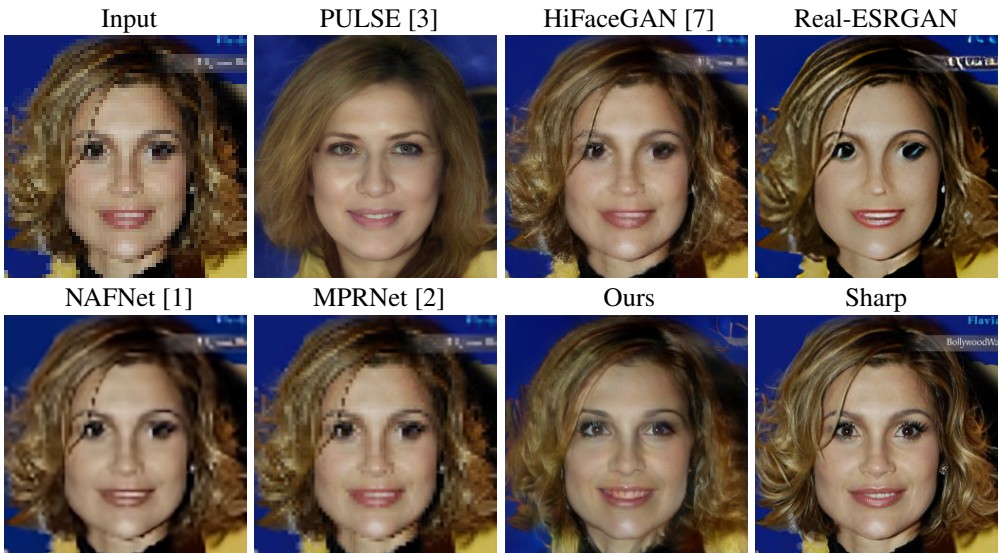

Figure 13: Additional super-resolution qualitative result on the CelebA-HQ dataset

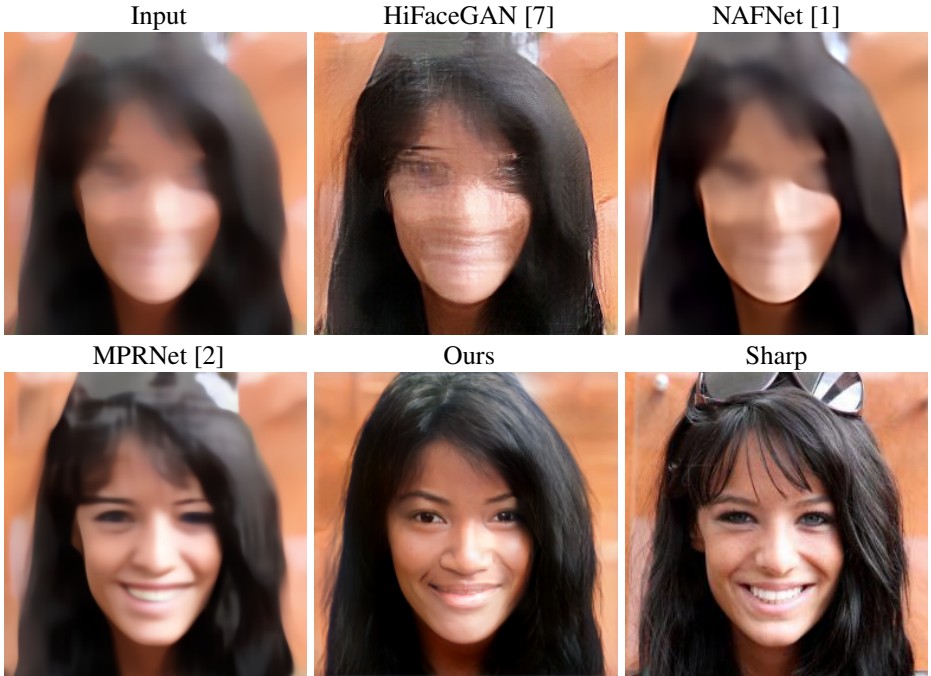

Figure 14: Additional image deblurring qualitative result on the CelebA-HQ dataset

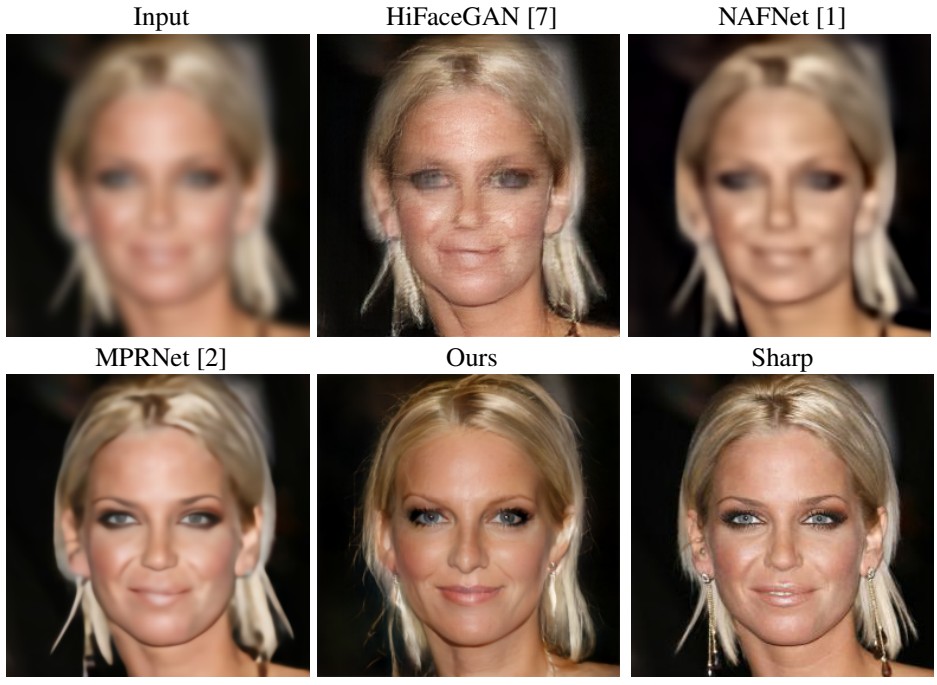

Figure 15: Additional image deblurring qualitative result on the CelebA-HQ dataset

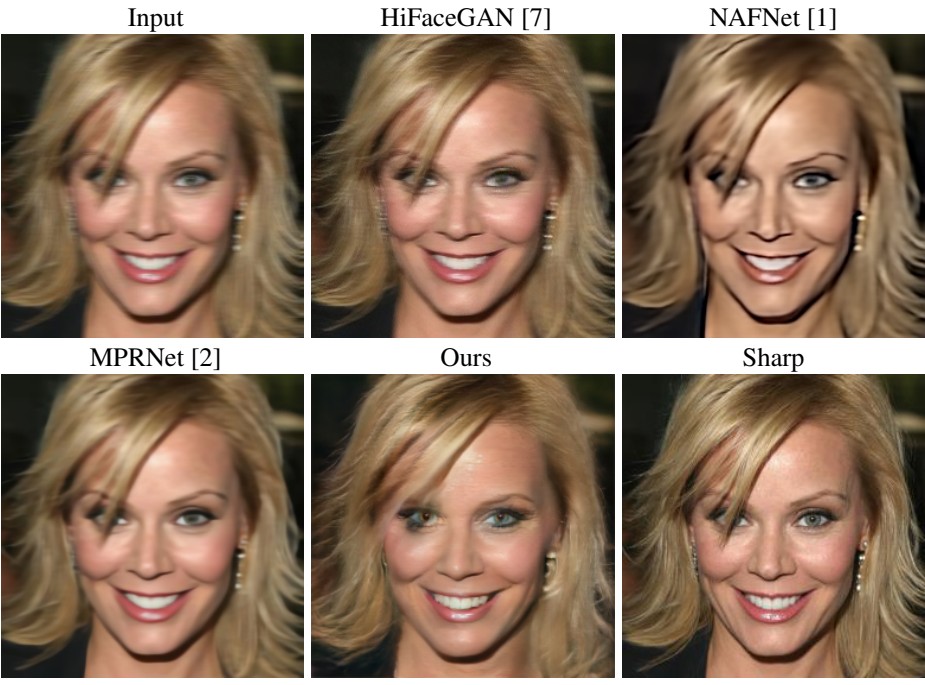

Figure 16: Additional image deblurring qualitative result on the CelebA-HQ dataset

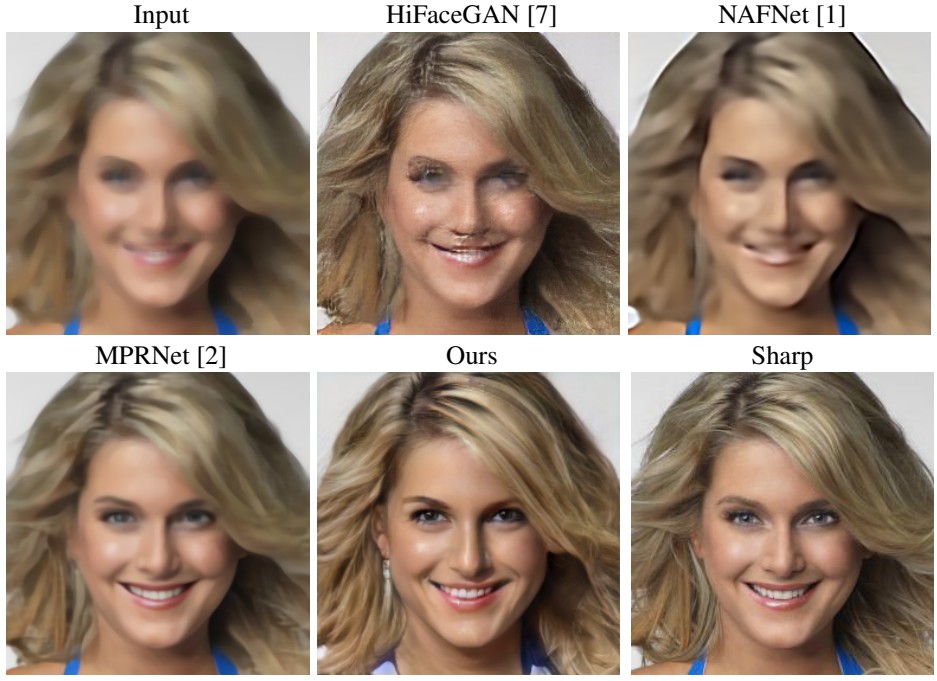

Figure 17: Additional image deblurring qualitative result on the CelebA-HQ dataset

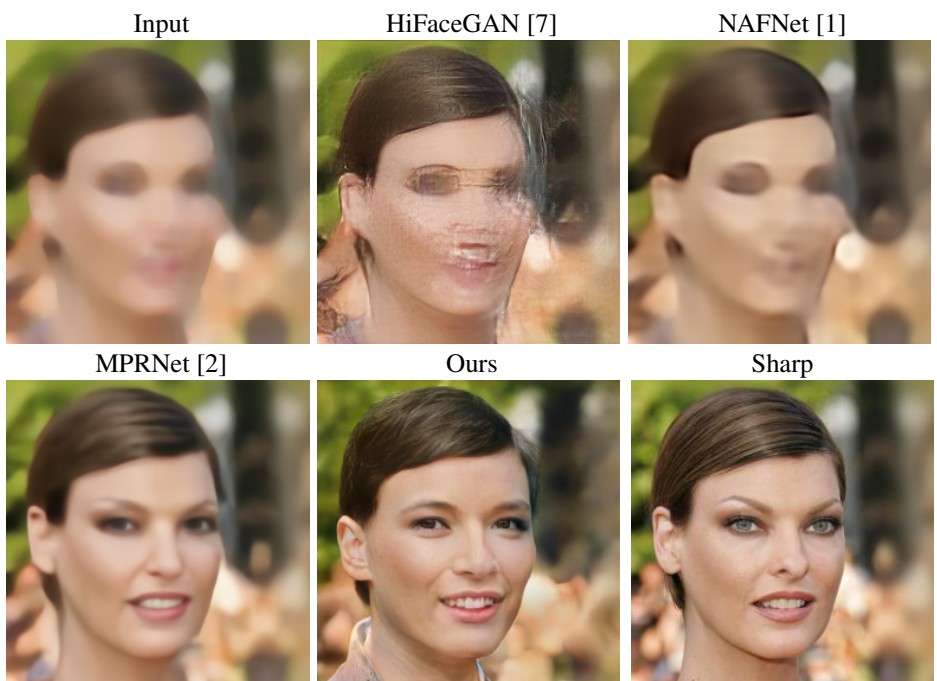

Figure 18: Additional image deblurring qualitative result on the CelebA-HQ dataset

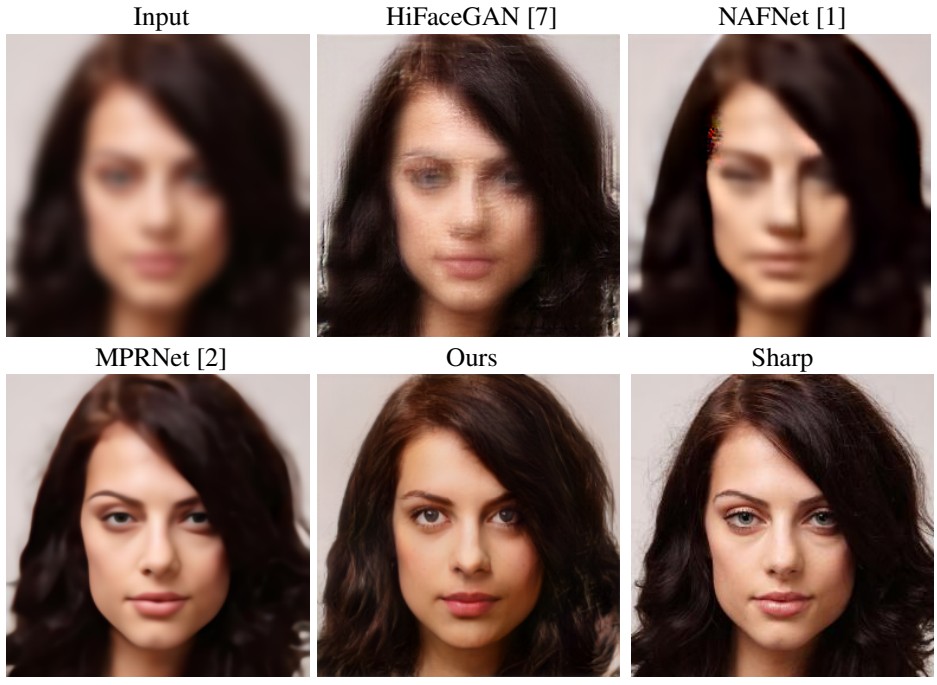

Figure 19: Additional image deblurring qualitative result on the CelebA-HQ dataset

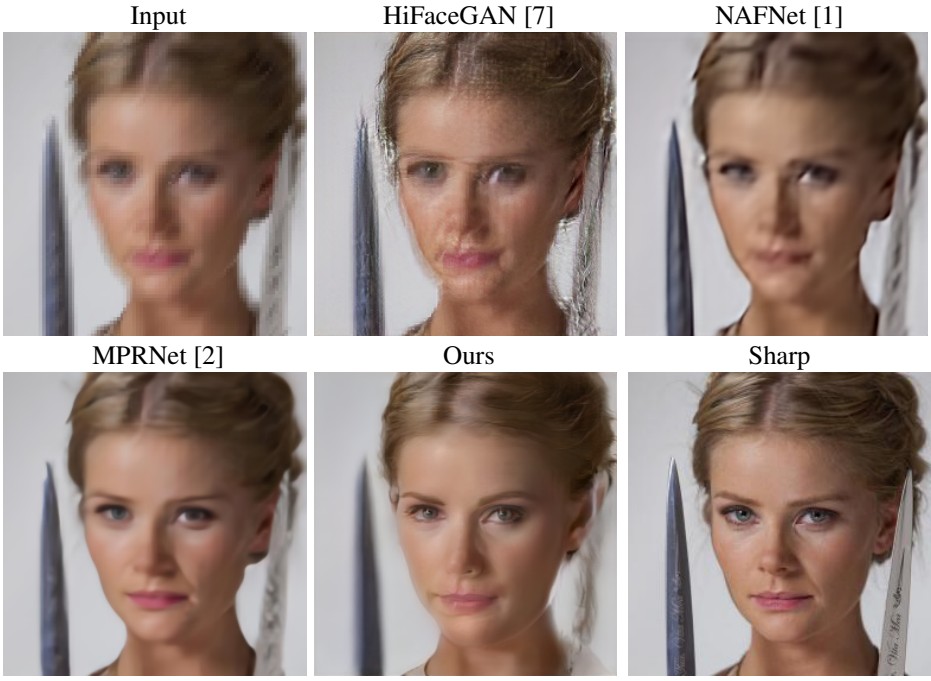

Figure 20: Additional multi-degradation image restoration result on the CelebA-HQ dataset

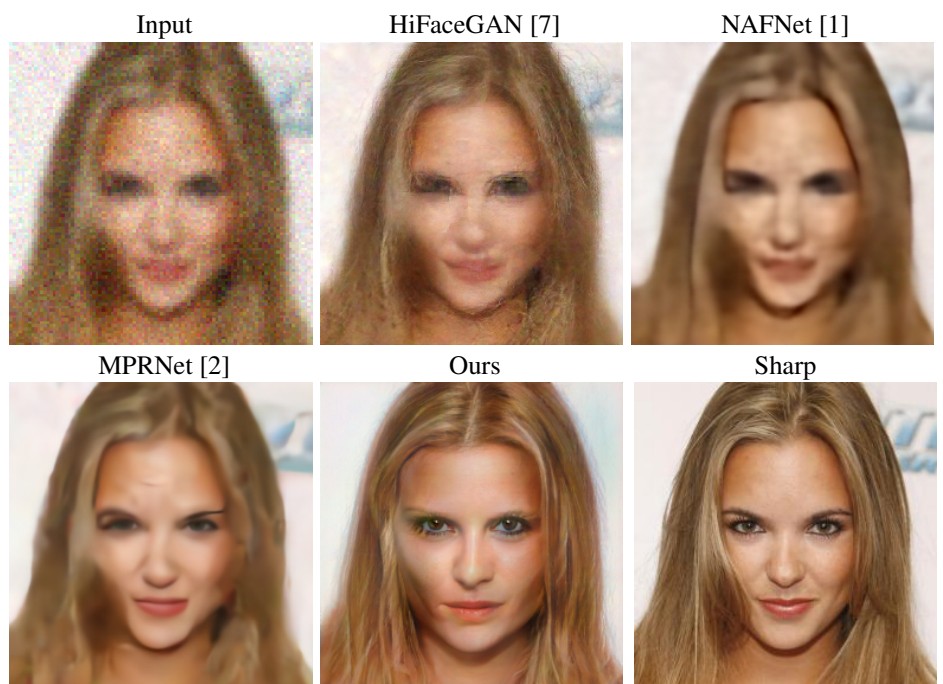

Figure 21: Additional multi-degradation image restoration result on the CelebA-HQ dataset

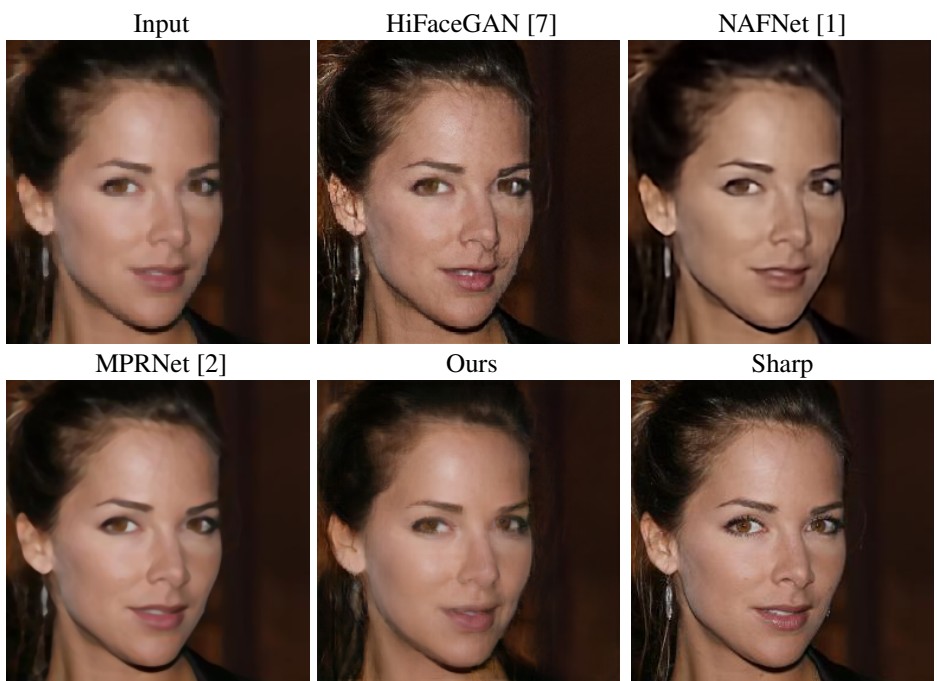

Figure 22: Additional multi-degradation image restoration result on the CelebA-HQ dataset

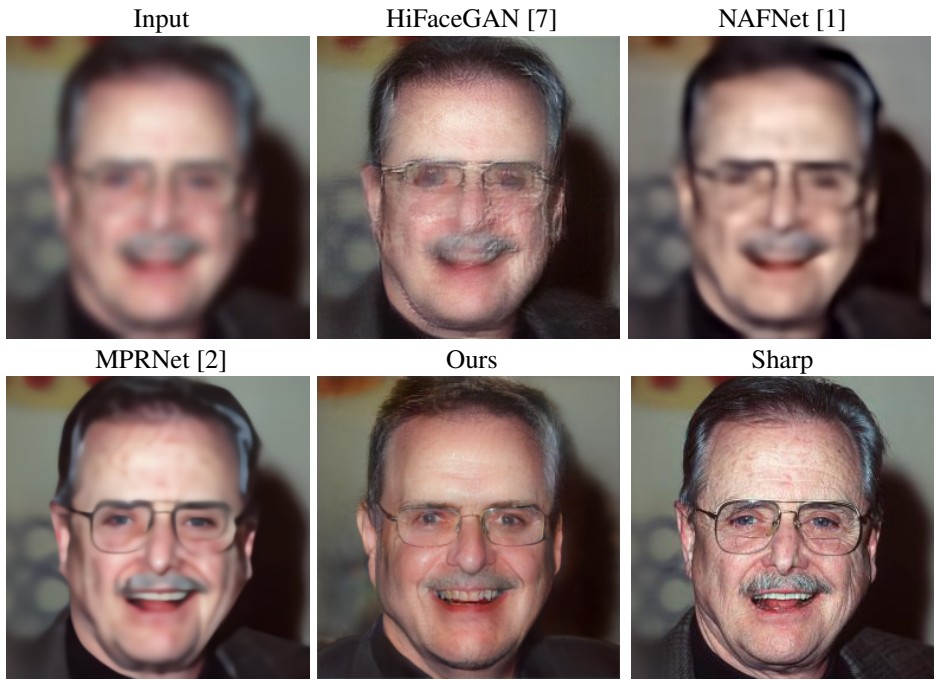

Figure 23: Additional multi-degradation image restoration result on the CelebA-HQ dataset

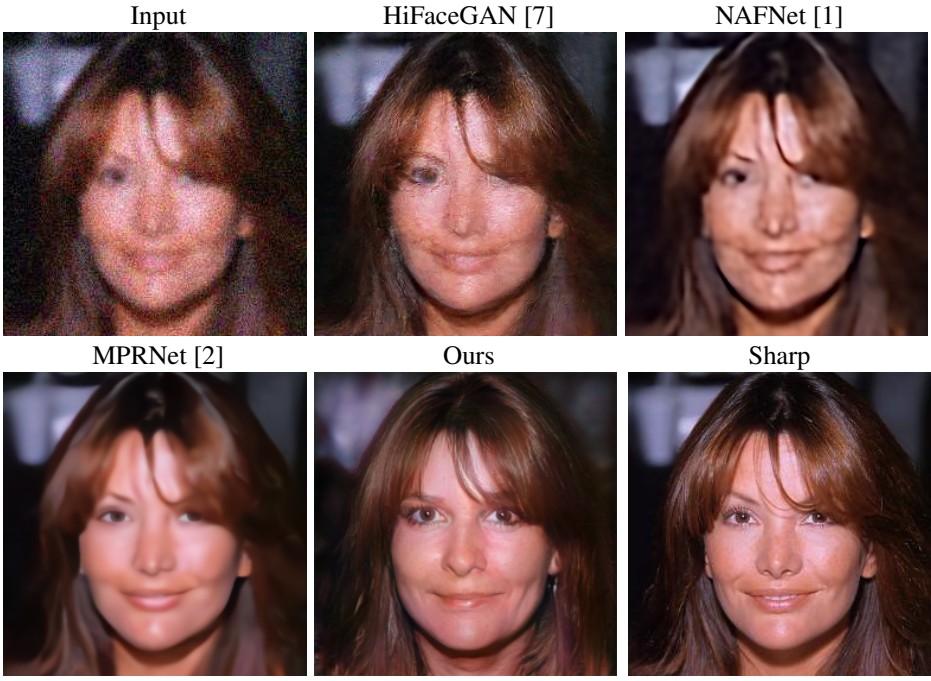

Figure 24: Additional multi-degradation image restoration result on the CelebA-HQ dataset

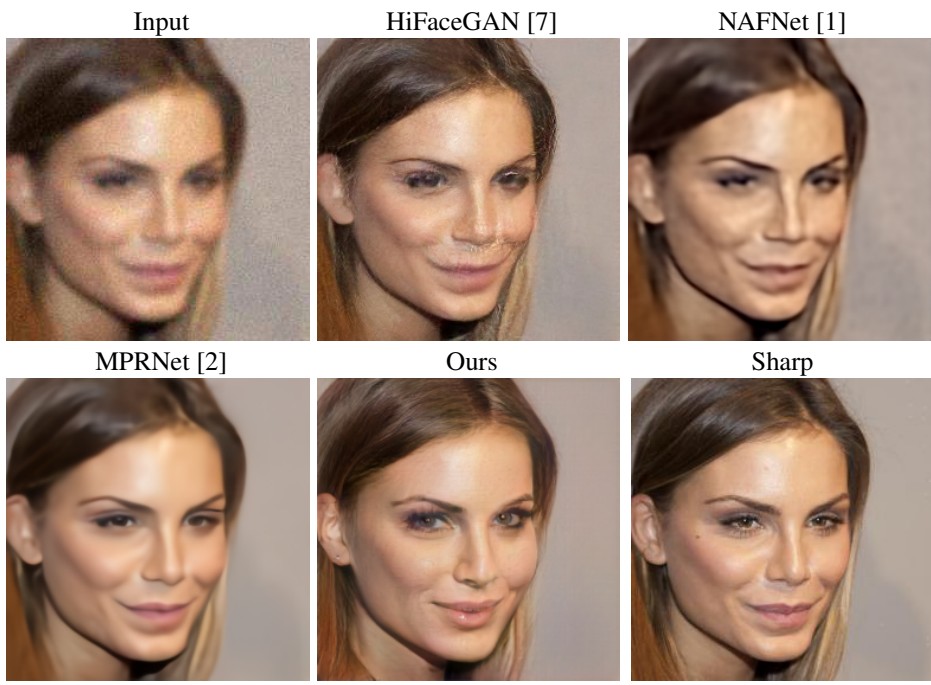

Figure 25: Additional multi-degradation image restoration result on the CelebA-HQ dataset

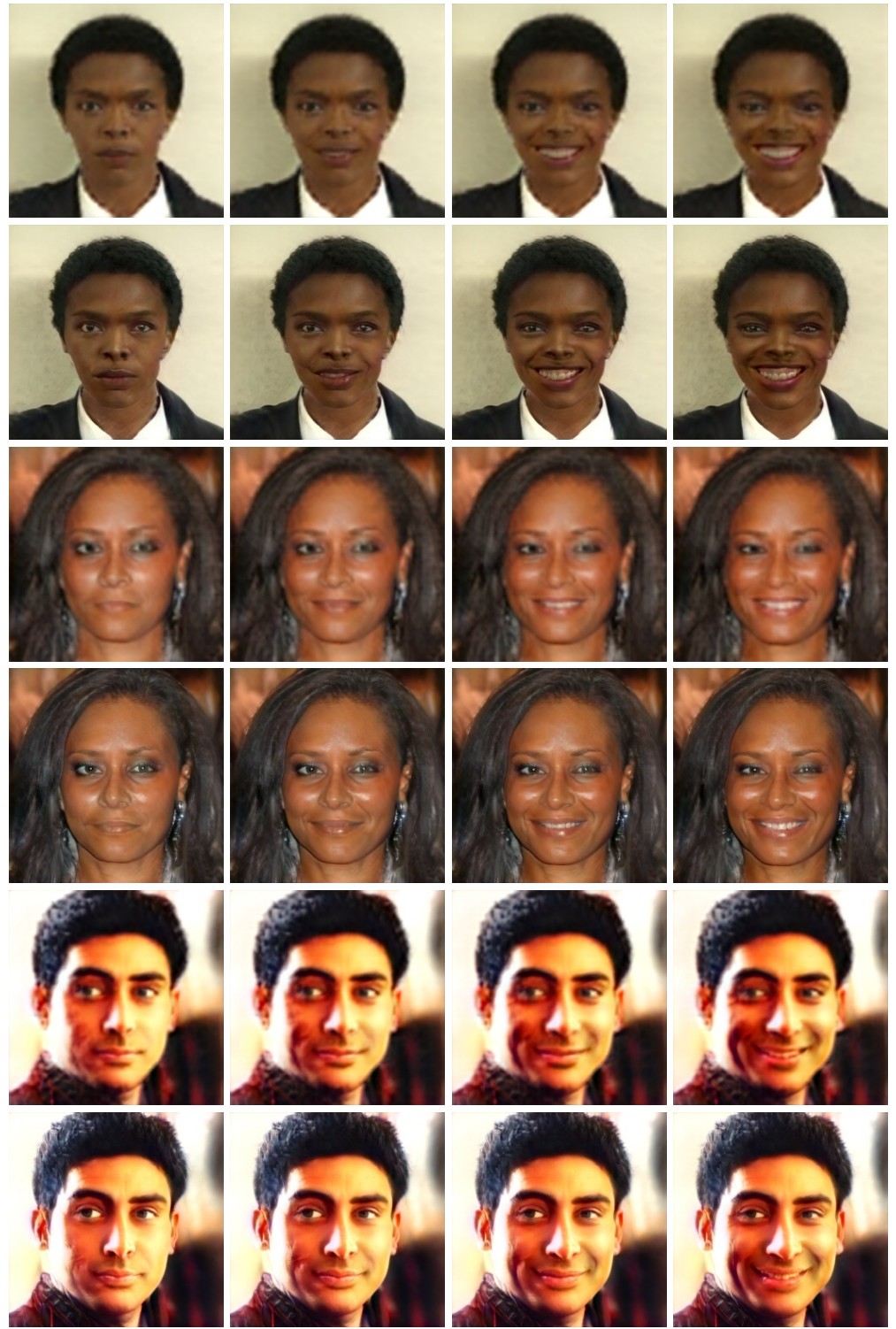

Figure 26: **GAN inversion and editing**. Each example is shown in two rows: For a given low-resolution image ($1^{st}$ image), we inverse it and apply the "smiling" editing direction on the latent code. The first row shows the edited images in the degraded domain, while the second row contains the edited images in the sharp domain.

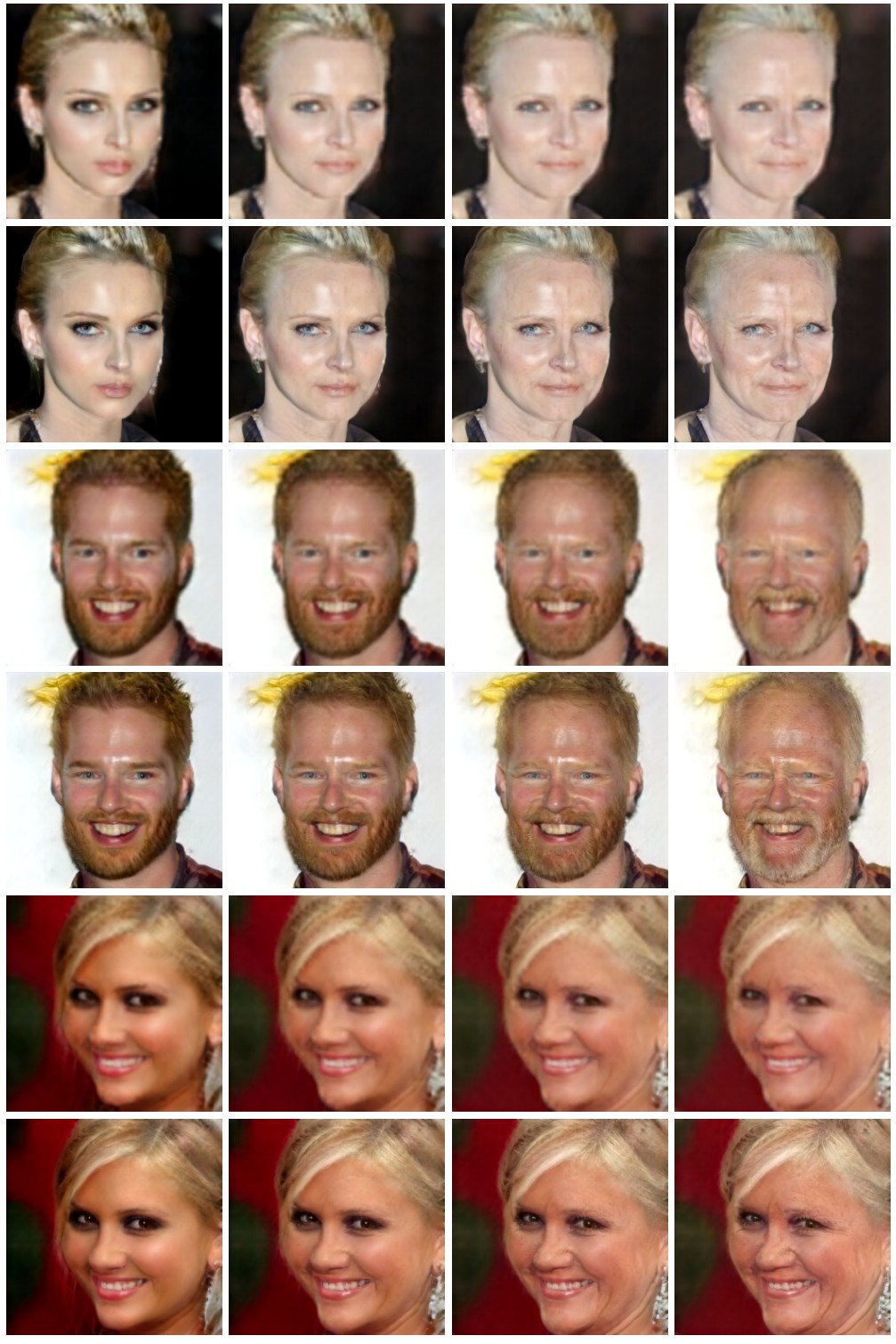

Figure 27: **GAN inversion and editing**. Each example is shown in two rows: For a given low-resolution image ($1^{st}$ image), we inverse it and apply the "aging" editing direction on the latent code. The first row shows the edited images in the degraded domain, while the second row contains the edited images in the sharp domain.

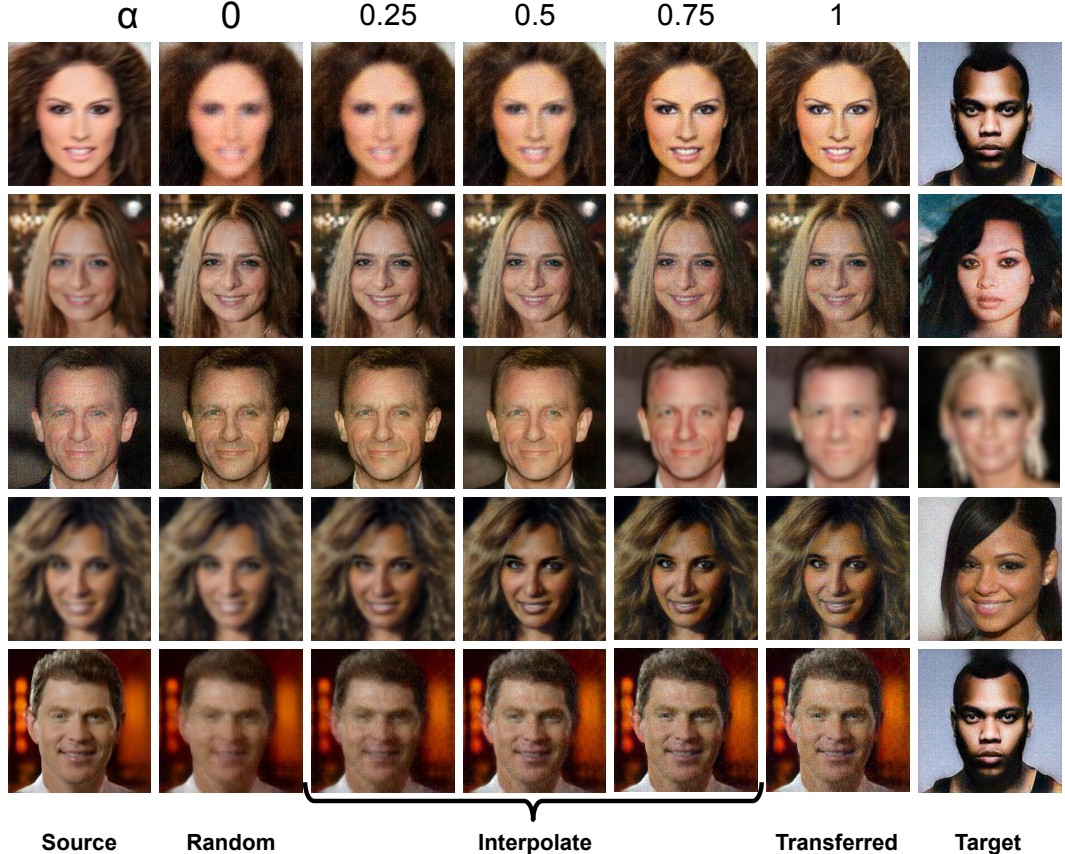

| α | 0 | 0.25 | 0.5 | 0.75 | 1 | |
|---|---|------|-----|------|---|---|
| Source | Random | | Interpolate | | Transferred | Target |

Figure 28: **Degradation synthesis.** In each row, from a source degraded image ($1^{st}$ column), we change its image degradation to a novel random one ($2^{nd}$ column) or copy the degradation from a reference image ($6^{th}$ column). We can also smoothly interpolate image degradations in-between the ones above, using an interpolation factor $\alpha \in [0, 1]$ ($3 - 5^{th}$ columns).

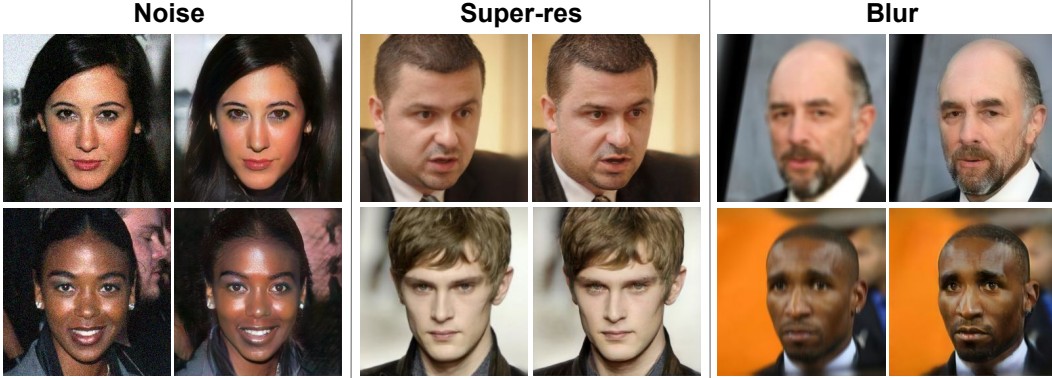

Figure 29: **Image restoration with small degradations.** For each degradation, we provide two examples from the CelebA-HQ dataset. Each example consists of the degraded input (left) and the restored image using our QC-StyleGAN. Best view in zoom.