# OpenReview forum: "QC-StyleGAN - Quality Controllable Image Generation and Manipulation"
_NeurIPS.cc/2022/Conference — NeurIPS 2022 Accept_

### Official Review · Reviewer_NzZ7 · 2022-07-07

**Rating:** 4
**Confidence:** 3
**Soundness:** 1 poor
**Presentation:** 2 fair
**Contribution:** 1 poor

**Summary:**

This paper aims to solve the problem that existing GAN methods are unfit for low-quality input images. The authors present a quality controllable image generation and manipulation method. The proposed method can be applied to many applications, like image restoration and image manipulation.

**Questions:**

1. Why this problem is important. Can existing inversion methods [7-13] solve this problem? Can the proposed method achieve better results than these inversion methods?

2. The authors use the term “a quality-control input” in Section 1, but explain it in Section 3.

3. There are so many arrows in Fig. 1, resulting in a confusing figure. Moreover, Fig. 1 does not appear in the text part of the paper.

4. My biggest concern for this paper is the validation part. In the experiments section, the authors show the results for image generation, image inversion, image editing, image restoration, image degradation, etc. However, for any application, only some visual results are shown, and no statistical performance values are reported. The details that use other methods to compute the results given in the paper are also missing. Are these methods fine-tuned using the corresponding dataset?

5. As shown in Table 1. the proposed method achieves a lower FID score for the degraded images than the sharp images on the FFHQ dataset, which is inconsistent with the results on the other datasets. Can the authors explain why?


**Limitations:**

The limitations are discussed.

**Strengths And Weaknesses:**

Strengths:
The proposed method is applied to many applications, like image restoration and image editing.

Weaknesses:
The writing of this paper needs to be improved. The paper is hard to follow. For example, there is no overview of the proposed method. Although there is an overview subsection, it is more like problem formulation. An overview in plain English and an illustration or framework are necessary before giving any formula.
The motivation, novelty, and contributions are not well described.
The method is not well-validated, and no extensive comparison results are given.

---

> ### Author Response · Authors · 2022-08-02
> **Our answers on your questions**
>
> We sincerely thank you for your valuable time and comments. We will address your concerns as follows:
>
> ### 1. Writing issues.
> Thanks for your comment. We have added a paragraph at the beginning of Section 3 to overview the structure of that section, including problem definition, the proposed network (QC-StyleGAN), and its related techniques. We have changed the title of Section 3.1 from "Overview" to "Problem definition". We provided the motivation for QC-StyleGAN in the Introduction. As for the system figures, Fig.2 illustrates the proposed QC-StyleGAN structure, while Fig. 1 summarizes its usage and the related techniques.
>
> ### 2. Can inversion methods [7-13] solve the problem? If yes, need comparisons.
> The inversion methods [2, 7-13, 29-40] are applicable to both QC-StyleGAN and the original StyleGAN. In fact, if we fix q=0 and use the same inversion method, the inversion results on QC-StyleGAN and on the original StyleGAN are almost identical.
> Among these inversion techniques, PTI [39] is state-of-the-art in terms of quality. Our proposed inversion is based on PTI with a small modification to guarantee the inversion result allows good image restoration. We already compare our inversion with PTI when comparing inversion results of QC-StyleGAN and StyleGAN2-Ada.
>
> ### 3. Quality control input term is used before its definition.
> In the Introduction section, we only provided a high-level introduction of QC-StyleGAN, and we mention the quality control input as a notable distinction of our proposed model. We gave the official definition of the quality control input when describing QC-StyleGAN structure in detail in Section 3.
>
> ### 4. Fig. 1 does not appear in the text part of the paper.
> Thanks for pointing out this writing issue. We have added a paragraph mentioning the figure in the revised manuscript’s Introduction section.
>
> ### 5. Lacks of extensive comparisons. Lacks quantitative experiments.
> We already provided quantitative evaluation on image generation (Table 1) and image inversion (PSNR scores in Section 4.3). We have added quantitative evaluation for Image editing and Image restoration in the sections E1 and E2 of the revised Supplementary. As for Degradation synthesis, we have added an experiment on degradation classification to confirm the correctness of our synthesized degradation in the sections E3 of the revised Supplementary. We will provide the details of these extra experiments in the next comment.
>
> ### 6. Missing details of other methods.
> For image restoration networks such as NAFNet and MPRNet, we re-train the models on our degradation images using their published code with default configuration. For GAN-based image restoration methods like HiFaceGAN and PULSE, we use their provided pre-trained models. We have added a few notes to Section 4.4 of the revised manuscript.
>
> ### 7. Why is the FID of degraded images higher than sharp ones, except in FFHQ dataset?
> We found that FID was not designed to evaluate degraded images; it focuses on texture of pretrained Image classes as discussed in paper [1]. Hence, FID score for degraded images may be noisy and unreliable. While we believe that a low FID score relatively indicates a good model, it is not reliable to compare FID scores of degraded-image datasets when the gap is not significant. It is even more defective when comparing FID of a sharp, clean dataset and the FID of a degraded one. Studying the behavior of the FID metric on these scenarios is an interesting topic for future research.
>
> [1]. Tuomas Kynkäänniemi, Tero Karras, Miika Aittala, Timo Aila, and Jaakko Lehtinen. The role of imagenet classes in frechet inception distance. arXiv preprint arXiv:2203.06026, 2022.

---

> > ### Author Response · Authors · 2022-08-02
> > **Extra quantitative experiments (Part 1)**
> >
> > Here we want to provide the details of the extra quantitative experiments we have conducted and added to Section E of the revised Supplementary:
> >
> > ### A. Image editing
> > We quantitatively evaluate the editability of our proposed GAN inversion method with QC-StyleGAN by measuring the amount of target attribute change with different editing magnitudes. Given a degraded image, we synthesize its corresponding edited versions using different editing magnitude $\gamma_t$ uniformly picked in the range of [-3, 3]. In this test, we choose the age editing direction. To measure the age change when editing, we leverage the off-the-shelf DEX VGG [1] model to estimate the face's age in the images. We perform this experiment on 100 multiple-degraded images and show the average age change for each magnitude in table below. We report results for both the manipulated degraded outputs and their sharp recovered version.
> >
> > | $\gamma$ | Our (degraded) | Ours (sharp) |
> > | :---               |    :----:   |         :---:    |
> > |-3.0       | 4.22                   | 15.04            |
> > |-2.33     | 3.39                   | 11.79            |
> > |-1.67     | 2.72                   | 9.16              |
> > |-1.0       | 1.86                   | 6.56              |
> > |-0.33     | 1.12                   | 3.5                |
> > |0.33      | 0.6                     | 0.72              |
> > |1.0        |-0.25                   |-1.78              |
> > |1.67      |-1.19                   |-4.24              |
> > |2.33      |-1.97                   |-6.12              |
> > |3.0        |-2.5                     |-7.63              |
> >
> > The age change scales consistently with the editing magnitude, confirming the editability of the latent space of our generator. Note that the age regression model was trained on sharp images; hence, it produces less significant changes on degraded images, which may not reflect the actual shift on these pictures.
> >
> > ### B. Quality-code-based Degradation Classification
> > As shown in Fig. 6 of the paper and Fig. 27 of the Supplementary, our degradation-transferred images have similar degradation as the target images. It confirms the high accuracy of our model as a degradation estimator. We further conducted a quantitative experiment in which our model was used as a degradation classification. First, we randomly generated 1000 facial images using our QC-StyleGAN and manually annotated them as having blur or not. Then, we used their quality codes and the labels above to train a linear classifier to predict whether a facial image is blurry or not based on the quality code. Finally, we use 100 facial images with blur and 100 facial images without blur to form a test set and test the accuracy of our classifier in blur classification. Our model gave an accuracy of **97.9%**, confirming the effectiveness of QC-StyleGAN as a degradation estimator.
> >
> > [1]. Rasmus Rothe, Radu Timofte, and Luc Van Gool. Dex: Deep expectation of apparent age from a single image. In Proceedings of the IEEE international conference on computer vision workshops, pages 10–15, 2015.

---

> > > ### Author Response · Authors · 2022-08-02
> > > **Extra quantitative experiments (Part 2)**
> > >
> > > ### C. Image restoration
> > > We quantitatively compare our proposed framework and the other image restoration methods on five common restoration tracks, including deblurring, super-resolution, denoising, JPEG removal, and multiple-degraded restoration. We use the CelebA-HQ dataset for this evaluation.
> > >
> > > Among the common evaluation metrics for this task, we found LPIPS more reliable and close to human perception. Therefore, we provide the LPIPS scores in the table below. Note that in the table, smaller LPIPS score means better. The **best** and *runner-up* values are marked in bold and italic, respectively. The mark '-' means the method is not applicable to the track.
> > > |                     | Blur      | Super-res. |   Noise  |  JPEG comp. | Multiple-deg |
> > > | :---               |    :----:   |         :---:    |      :---:   |       :---:          |          :---:      |
> > > | HiFaceGAN | 0.216   | *0.125*      | *0.126* | **0.053**        | 0.364           |
> > > |ESRGAN      | -           | 0.148        |     -        | -                     |    -                |
> > > |DnCNN        | -           |    -             |**0.080**|  -                    |    -                |
> > > |MPRNet      |**0.194**| 0.230       | 0.143     | 0.128             | *0.299*         |
> > > |PULSE        | -            | 0.296       | -             | -                     | -                   |
> > > |mGANPrior | -            | 0.265        | -            | -                     | -                   |
> > > |GLEAN       | -            |**0.072**   | -           |   -                    | -                    |
> > > |Ours           | *0.195* | 0.177        | 0.183    | *0.118*           |**0.260**       |
> > >
> > > Although our method was not tailored to handle this restoration task specifically, it performed reasonably well and outperformed many baseline methods in each task. Particularly, QC-StyleGAN privided the best LPIPS score when having multiple degradations in the input images.  Also, we found that one degraded image may correspond to multiple possible sharp images. Our restoration results sometimes looked reasonable but did not match the ground-truth ones, severely hurting our LPIPS scores.
> > >
> > > To avoid the mismatching ground-truth issue, we also use NIQE [1], which is a no-reference image quality metric, and report the scores in the table below. Again, smaller score means better. The **best** and *runner-up* values are marked in bold and italic, respectively.
> > > |                     | Blur      | Super-res. |   Noise  |  JPEG comp. | Multiple-deg |
> > > | :---               |    :----:   |         :---:    |      :---:   |       :---:          |          :---:      |
> > > |HiFaceGAN  | *5.95*  | **5.32**     | *6.01*    | *4.95*            | *5.916*         |
> > > |ESRGAN      | -          | 6.35           | -             | -                    | -                   |
> > > |DnCNN         | -          | -                 | 6.93       | -                   | -                    |
> > > |MPRNet        | 8.12   | 6.73           | 7.33        | 7.64             | 8.97              |
> > > |PULSE          | -         | 6.29           | -             | -                   | -                    |
> > > |mGANPrior   | -         | 6.02           | -             | -                   | -                    |
> > > |GLEAN         | -         | 7.29           | -             | -                   | -                |
> > > |Ours             |**5.83**|*5.45*|**5.41**|**4.51**|**5.64**|
> > >
> > > QC-StyleGAN provides the best NIQE score in nearly all tracks. It confirms that our image restoration can produce the highest output quality in terms of naturalness [1] while still maintaining comparable perceptual similarity [2] compared to the competitors.
> > >
> > > [1]. Anish Mittal, Rajiv Soundararajan, and Alan C Bovik. Making a “completely blind” image quality analyzer. IEEE Signal processing letters, 20(3):209–212, 2012.
> > >
> > > [2]. Richard Zhang, Phillip Isola, Alexei A Efros, Eli Shechtman, and Oliver Wang. The unreasonable effectiveness of deep features as a perceptual metric. In CVPR, 2018.

---

### Official Review · Reviewer_EPGp · 2022-07-10

**Rating:** 6
**Confidence:** 4
**Soundness:** 3 good
**Presentation:** 3 good
**Contribution:** 3 good

**Summary:**

StyleGAN is able to generate high-resolution images and produce semantically disentangled and smooth latent space at the same time. This paper proposed an extension of StyleGAN that is able to control the quality of the generated images by giving a controllable vector. This extension not only allows to generate degraded images but also brings some interesting applications such as image editing and image restoration by applying GAN inversion and manipulation techniques. Although there are not many quantitative comparisons, qualitative results show impressive effectiveness.

**Questions:**

1. I didn't find any main text mentioned the Figure 1.
2. I noticed that in Table 1, the FID score of LSUN Church of the degraded image generation model is relatively higher than others. Do you have any explanation for it?

**Limitations:**

As mentioned in my second question, the proposed method may not perform well in some cases. It is important to investigate the reason and it should be discussed in the paper. It will be beneficial to the readers and may help readers to conduct further research in the future.

**Strengths And Weaknesses:**

Strengths:
The idea is easy to understand and I think maybe the concept can be extended to different research. The qualitative results show impressive effectiveness.

Weaknesses:
There is no baseline method of the degraded image generation models for comparison. Although the qualitative results are good, it would be more convincing if there are more quantitative comparisons.

---

> ### Author Response · Authors · 2022-08-02
> **Our answers on your questions**
>
> We sincerely thank you for your valuable time and comments. We will address your concerns as follows:
>
> ### 1. There is no baseline method of the degraded image generation.
> We can use StyleGAN2-Ada  models trained on degraded images as baseline methods. Here is the FID comparison between these baseline models and our QC-StyleGAN on degraded images:
> | Method               | FFHQ | AFHQ Cat | LSUN Church |
> | :---                      | :---:     | :---:            | :---:                  |
> | StyleGAN2-Ada  | 4.38   | 4.70           | 5.16               |
> | QC-StyleGAN    | 3.23    | 3.91           | 4.58               |
>
> ### 2. No text mentioned Figure 1.
> Thanks for pointing out this writing issue. We have added a paragraph mentioning the figure in the revised manuscript’s Introduction section.
>
> ### 3. Why is the FID score of LSUN Church of the degraded image generation model relatively higher than others in Table 1?
> - We found that FID was not designed to evaluate degraded images; it focuses on texture of pretrained Image classes as discussed in paper [1]. Hence, FID score for degraded images may be noisy and unreliable. While we believe that a low FID score relatively indicates a good model, it is not reliable to compare FID scores of degraded-image datasets when the gap is not significant.
> - Even if we assume FID scores are reliable, we can explain this as the nature of the dataset. As can be seen in the table above, FID score of the StyleGAN2-Ada model on degraded LSUN-Church images is 5.16, which is much higher than the FID score of the StyleGAN2-Ada model on the clean LSUN-Church images (3.86) as well as the other models. Hence, the high FID score comes from the data itself. We theorize that StyleGAN’s modeling capacity is good enough to fit the degraded images on simple domains like faces but not enough to fit the degraded images on complex domains like churches.

---

### Official Review · Reviewer_H13T · 2022-07-11

**Rating:** 5
**Confidence:** 5
**Soundness:** 3 good
**Presentation:** 4 excellent
**Contribution:** 2 fair

**Summary:**

This paper targeted a very interesting problem. When we use a GAN-inversion method to edit images, the GAN can only generate sharp images, and what if we got a degraded image and we don't want its degradation to be distorted? This paper gives a method to solve this problem. In addition to an original StyleGAN network, this paper proposes a DegradBlock to add degradation that is controlled by an additional degradation quality code. When a new image comes, the method not only inverts the sharp image but also inverts its degradation. The results show good performance and several applications are studied.


**Questions:**

I hope the author responds to the weaknesses I raised above and the following additional questions:

1. Estimating the degradation of an image is a difficult task. From the examples shown, most images are badly degraded. Will the proposed method perform well on small degradations? such as small noise, blurring with small sigma, and jpeg with a quality score of 70?
2. How good is it as a degradation estimator? Can we estimate degradation using the proposed method?
3. Are there big differences between the two inversion results?
4. How good is the restoration performance? i.e. compare with GLEAN [1] or mGANprior [2] (BTW these works should be cited)


=================update:
After the reviewer-author discussion, I am willing to increase my rating from 4 to 5.


**Ethics Review Area:**

["I don’t know"]

**Limitations:**

This work is related to face images but I guess there is no obvious potential negative societal impact.
The limitation is not discussed.

**Strengths And Weaknesses:**

Strengths:
1. The question of "what if we got a degraded image" is interesting.
2. It is the first GAN method that can be controlled to generate degraded images
3. Nice performance.
4. A Generator that knows different degradation is a good idea for blind image restoration.

Weaknesses:
1. Although interesting, the practice value of the studied problem is not that obvious. Given the mentioned setting, why would people want a degraded image? Maybe the restoration task is more valuable.
2. The design of the DegradBlock lacks novelty. It is like an AddIn controlled by the quality code and with some conv layers. The direct product operation makes it give 0 when the quality code is 0. This method doesn't give us much insight technically. Although I comment on the novelty, it is the last reason that affects my rating.
3. The proposed method can be used as a blind Face Restoration method, but the provided experiments are not enough to show the actual potential.

---

> ### Author Response · Authors · 2022-08-02
> **Our answers on your questions**
>
> We sincerely thank you for your valuable time and comments. We will alleviate your concerns as follows:
>
> ### 1. Why would people want a degraded image? Maybe the restoration task is more valuable.
> Thanks for your question. Although restored images look great, they are not always desirable:
> -  In many scenarios, only a part of the data is manipulated, and the manipulated result should have consistent quality as the rest. One scenario is to edit a few frames in a video. Another scenario is when the manipulated input is only an image crop from a large image, and the editing result must be added back to the original image. A typical example is when editing a facial image, we often extract an aligned crop around the face, apply GAN-based editing, and plug the manipulated crop into the full image.
> - Restored images often have bias, which may not be desirable. For example, facial image superresolution/deblurring often introduces identity bias.
> We have added the first reason to our revised paper's Introduction section.
>
> ### 2. The design of the DegradBlock lacks novelty. It is like an AddIn controlled by the quality code and with some conv layers.
> The design of DegradBlock was actually inspired by PCA, and its mechanism is very different from AdaIn. In DegradBlock, we first use a convolution layer c to predict D_q principal components r_1^{(i)}, then compute their linear combination with q as the component weights. In AdaIn, the feature map is first normalized, then scaled and shifted channel-wise; the channels are processed independently with no grouping or linear combination. We tried to replace DegradBlocks with AdaIn-like blocks, and the FID for clean and degraded images on the FFHQ dataset are 4.41 and 5.28, respectively, which are much higher. It confirms the superiority of our proposed design. We have added the discussions to Section 3.2 and Section 4.6 in our revised manuscript.
>
> ### 3. Will the method perform well on small degradations?
> Our QC-StyleGAN models were trained to handle image degradations at various degrees, unlike many deep-learning-based image restoration techniques. We provide some image restoration results on CelebA-HQ images under small degradations, using our QC-StyleGAN model trained on the FFHQ dataset, in Section D5 of our revised Supplementary as well as in this anonymous link [http://bit.ly/3oPOGZk](http://bit.ly/3oPOGZk) (anonymous link). As can be seen, these images are still recovered effectively.
>
> ### 4. Can we estimate degradation using the proposed method?
> As shown in Fig. 6 of the paper and Fig. 27 of the Supplementary, our degradation-transferred images have similar degradation as the target images. It confirms the high accuracy of our model as a degradation estimator. We further conducted a quantitative experiment in which our model was used as a degradation classification. First, we randomly generated 1000 facial images using our QC-StyleGAN and manually annotated them as having blur or not. Then, we used their quality codes and the labels above to train a linear classifier to predict whether a facial image is blurry or not based on the quality code. Finally, we use 100 facial images with blur and 100 facial images without blur to form a test set and test the accuracy of our classifier in blur classification. Our model gave an accuracy of **97.9%**, confirming the effectiveness of QC-StyleGAN as a degradation estimator.
>
> ### 5. Are there big differences between the two inversion results?
> We assume your question is about the differences between our inversion results and StyleGAN2-Ada inversion ones:
> - First, our reconstructed images can be easily converted to their sharp version. In contrast, inversion with StyleGAN2-Ada only gives us fitted degraded images.
> - Moreover, since QC-StyleGAN models both sharp and degraded images, the inversion results often stay in-distribution, allowing good editing results. In contrast, the original StyleGAN2-Ada network only models sharp images; editing its inversion on degraded inputs may lead to unrealistic outcomes. We provide the comparison between their manipulation results in Section F1 of our revised Supplementary as well as in this anonymous link [http://bit.ly/3zu81DY](http://bit.ly/3zu81DY).

---

> > ### Author Response · Authors · 2022-08-02
> > **Image restoration performance**
> >
> > ### 6. How good is the restoration performance? Also, compare with GLEAN or mGANprior.
> > We quantitatively compare our proposed framework and the other image restoration methods on five common restoration tracks, including deblurring, super-resolution, denoising, JPEG removal, and multiple-degraded restoration. We use the CelebA-HQ dataset for this evaluation. Besides the competitors mentioned in the paper, we also evaluate GLEAN and mGANPrior as your suggestion.
> >
> > Among the common evaluation metrics for this task, we found LPIPS more reliable and close to human perception. Therefore, we provide the LPIPS scores in the table below. Note that in the table, smaller LPIPS score means better. The **best** and *runner-up* values are marked in bold and italic, respectively. The mark '-' means the method is not applicable to the track.
> > |                     | Blur      | Super-res. |   Noise  |  JPEG comp. | Multiple-deg |
> > | :---               |    :----:   |         :---:    |      :---:   |       :---:          |          :---:      |
> > | HiFaceGAN | 0.216   | *0.125*      | *0.126* | **0.053**        | 0.364           |
> > |ESRGAN      | -           | 0.148        |     -        | -                     |    -                |
> > |DnCNN        | -           |    -             |**0.080**|  -                    |    -                |
> > |MPRNet      |**0.194**| 0.230       | 0.143     | 0.128             | *0.299*         |
> > |PULSE        | -            | 0.296       | -             | -                     | -                   |
> > |mGANPrior | -            | 0.265        | -            | -                     | -                   |
> > |GLEAN       | -            |**0.072**   | -           |   -                    | -                    |
> > |Ours           | *0.195* | 0.177        | 0.183    | *0.118*           |**0.260**       |
> >
> > Although our method was not tailored to handle this restoration task specifically, it performed reasonably well and outperformed many baseline methods in each task. Particularly, QC-StyleGAN privided the best LPIPS score when having multiple degradations in the input images.  Also, we found that one degraded image may correspond to multiple possible sharp images. Our restoration results sometimes looked reasonable but did not match the ground-truth ones, severely hurting our LPIPS scores.
> >
> > To avoid the mismatching ground-truth issue, we also use NIQE [1], which is a no-reference image quality metric, and report the scores in the table below. Again, smaller score means better. The **best** and *runner-up* values are marked in bold and italic, respectively.
> > |                     | Blur      | Super-res. |   Noise  |  JPEG comp. | Multiple-deg |
> > | :---               |    :----:   |         :---:    |      :---:   |       :---:          |          :---:      |
> > |HiFaceGAN  | *5.95*  | **5.32**     | *6.01*    | *4.95*            | *5.916*         |
> > |ESRGAN      | -          | 6.35           | -             | -                    | -                   |
> > |DnCNN         | -          | -                 | 6.93       | -                   | -                    |
> > |MPRNet        | 8.12   | 6.73           | 7.33        | 7.64             | 8.97              |
> > |PULSE          | -         | 6.29           | -             | -                   | -                    |
> > |mGANPrior   | -         | 6.02           | -             | -                   | -                    |
> > |GLEAN         | -         | 7.29           | -             | -                   | -                |
> > |Ours             |**5.83**|*5.45*|**5.41**|**4.51**|**5.64**|
> >
> > QC-StyleGAN provides the best NIQE score in nearly all tracks. It confirms that our image restoration can produce the highest output quality in terms of naturalness [1] while still maintaining comparable perceptual similarity [2] compared to the competitors.
> >
> > These experiments and discussions were added into Section E2 of our revised Supplementary.
> >
> > [1]. Anish Mittal, Rajiv Soundararajan, and Alan C Bovik. Making a “completely blind” image quality analyzer. IEEE Signal processing letters, 20(3):209–212, 2012.
> >
> > [2]. Richard Zhang, Phillip Isola, Alexei A Efros, Eli Shechtman, and Oliver Wang. The unreasonable effectiveness of deep features as a perceptual metric. In CVPR, 2018.

---

> > > ### Comment · Reviewer_H13T · 2022-08-02
> > > **Response to Paper 7688 authors**
> > >
> > > Thanks for your reply!
> > >
> > > I am satisfied with the author's answers to most of my questions. But I still have concerns about the fourth and fifth questions.
> > >
> > > 4. Distinguishing blurred and unblurred images may be too simple for degradation estimation. I can even simply use a gradient distribution for classification. Maybe the author can try to divide the degree of blurring into 5 to 10 levels, or the degree of noise into several levels, and then study the accuracy of the degradation estimation.
> > >
> > > 5. I mean to conduct the proposed method for the same input image multiple times. Is the proposed method stable enough to output almost the same editing result every time? Since the authors mentioned that the proposed method might be used to edit video, it would appear flickering if the result differs from frame to frame. So the stability of the method is very important. This requires quantitative analysis.
> > >
> > > Looking forward the authors' reply.

---

> > > > ### Author Response · Authors · 2022-08-05
> > > > **Our answers on your questions**
> > > >
> > > > Thanks for your interesting questions. Please find our answers below:
> > > >
> > > > ### 4. Accuracy of the degradation estimation when dividing the degree of blurring into 5 to 10 levels.
> > > > When we divide the degree of blurring into 5 levels, the blur classification accuracy is *85%*. When we divide the degree of blurring into 10 levels, the blur classification accuracy is *77%*. These results are pretty good since we find the difference between consecutive blur levels is very subtle and hard for humans to distinguish (two examples are provided at [https://bit.ly/3zZHGzi](https://bit.ly/3zZHGzi) and [https://bit.ly/3P3pbhN](https://bit.ly/3P3pbhN)).
> > > >
> > > > ### 5. Is the proposed method stable enough to output almost the same editing result every time?
> > > > Thanks for your question. We have conducted experiments to verify the stability of our inversion and editing on degraded CelebA-HQ inputs images. We tried the editing magnitudes &pm;3 for 3 editing tasks on gender, age, and smiling. For each input image and each editing operator, we executed the operator 3 times to get 3 manipulated outputs and compare them pairwise using the PSNR and LPIPS metrics. The manipulated degraded images are pretty similar, with the PSNR score *44.42 &pm; 2.71* and the LPIPS score *0.004 &pm; 0.003*. When recovering the sharp version of these images, the PSNR score is still high (*41.84 &pm; 2.07*) and the LPIPS score is still very good (*0.006 &pm; 0.005*).
> > > >
> > > > We generated some quantitative videos and provided them in this anonymous link: [https://bit.ly/3JEIz3C](https://bit.ly/3JEIz3C). For each video, we show the same manipulation in 6 runs. As can be seen, our manipulated results are quite stable, with minor flickers mainly appearing on the background or the hair region. If we mask out the background and keep only the face region (the hair region is still kept), the scores in the previous experiments get improved: For degraded images, the PSNR score is *47.18 &pm; 2.27*, and the LPIPS score is *0.0013 &pm; 0.0006*. For sharp-recovered images, the PSNR score is *43.67 &pm; 2.28*, and the LPIPS score is *0.0028 &pm; 0.0013*. It confirms that our inversion and editing are pretty stable on the target object.

---

> > > > > ### Comment · Reviewer_H13T · 2022-08-05
> > > > > **Response to Paper 7688 authors (Second Round)**
> > > > >
> > > > > Thanks for the quick answer.
> > > > >
> > > > > I am generally feeling good about the questions now. Would these results appear in the final paper? The current version is still carrying redundant parts. More content will enrich the perception and value of this paper. If the final version contains enough discussion, the novelty issue of this paper are less severe, and I will increase my rating.
> > > > >
> > > > > Thanks

---

> > > > > > ### Author Response · Authors · 2022-08-06
> > > > > > **We have uploaded a revision addressing your request.**
> > > > > >
> > > > > > We sincerely thank you for your positive comments and suggestions.
> > > > > >
> > > > > > We have uploaded a revision. In the updated manuscripts, we include extra results and discussions, such as the quantitative experiments for image restoration, the quality-code-based degradation classification experiments, and more. Due to the space limit, some experiments, such as the stability test and the image restoration results on small degradations, are added to the revised Supplementary.
> > > > > >
> > > > > > We hope this revision contains enough content as you requested.

---

> > > > > > > ### Comment · Reviewer_H13T · 2022-08-07
> > > > > > > **Thanks for your revision!**
> > > > > > >
> > > > > > > Thanks to the authors for their efforts. Given the current revision, I am willing to raise my score to weak accept (6).
> > > > > > >
> > > > > > > But by the way, I checked the current revision. I think there is still work to do to incorporate some experiments into the main text. After all, the author's use of space is extravagant in the current version, such as tables and pictures :)

---

> > > > > > > > ### Author Response · Authors · 2022-08-07
> > > > > > > > **Thank you for your positive score and valuable feedbacks!**
> > > > > > > >
> > > > > > > > Thanks for your updated score. We appreciate your valuable comments, which greatly help to improve the quality of our paper.

---

### Official Review · Reviewer_mDFp · 2022-07-15

**Rating:** 7
**Confidence:** 2
**Soundness:** 3 good
**Presentation:** 3 good
**Contribution:** 3 good

**Summary:**

The introduction of high-quality image generation models, particularly the Style-GAN family, provides a powerful tool to synthesize and manipulate images. However, existing models are built upon high-quality (HQ) data as desired outputs, making them unfit for in-the-wild low-quality (LQ) images, which are common inputs for manipulation. In this work, the authors bridge this gap by proposing a novel GAN structure that allows for generating images with controllable quality. The network can synthesize various image degradation and restore the sharp image via a quality control code. The proposed QC-StyleGAN can directly edit LQ images without altering their quality by applying GAN inversion and manipulation techniques

**Questions:**

The reviewer is not very familiar with the scope of this paper. There are no more questions.

**Ethics Review Area:**

["I don’t know"]

**Limitations:**

The reviewer is not very familiar with the scope of this paper. The reviewer has no idea of the limitations.

**Strengths And Weaknesses:**

The proposed QC-StyleGAN is novel and can generate both clean and degraded images with a quality-control input.
The QC-StyleGAN allows more accurate GAN inversion on low-quality images, making image manipulation applicable on these inputs. It provides an efficient GAN-based image restoration solution. QC-StyleGAN also supports novel image degradation synthesis tasks.

The reviewer think that the proposed method is novel and recommend to accept the paper.

---

> ### Author Response · Authors · 2022-08-03
> **Thanks for your comments**
>
> We sincerely thank you for your comments and good rating. We are encouraged by your positive comments on our novelty, our well-performed model, and its diverse applications.

---

### Author Response · Authors · 2022-08-02
**Thanks for your comments. The revision is uploaded.**

We thank the reviewers for the detailed comments. We will post our answer to each reviewer separately.

Besides, according to the rules of NeurIPS this year, we have uploaded a revised manuscript and a revised supplementary PDF. In these documents, we highlight the modified text to address reviewers’ comments in blue color.

---

### Meta-Review · Area_Chair_fhqZ · 2022-08-30

**Recommendation:** Accept
**Confidence:** Certain

**Metareview:**

QC-StyleGAN provides a controllable way of generating images with certain types of corruption. Generating corrupted images has many applications because they are common in real-world situations, so having the ability to generate samples from distributions with controllable corruptions is a powerful idea. The reviewers generally felt the paper should be accepted, although the highest score came from the least confident reviewer. The overall average is still on the accept side, and I feel the paper has enough interesting ideas and potential applications to be useful to the NeurIPS audience, so I recommend acceptance.

**Award:**

No

---

### Decision · Program_Chairs · 2022-09-14

Accept